# Advancing Model Pruning via Bi-level Optimization

**Yihua Zhang**[1,*]   **Yuguang Yao**[1,*]   **Parikshit Ram**[2]   **Pu Zhao**[3]   **Tianlong Chen**[4]

**Mingyi Hong**[5]       **Yanzhi Wang**[3]       **Sijia Liu**[1,2]

[1]Michigan State University, [2] IBM Research, [3]Northeastern University,
[4]University of Texas at Austin, [5]University of Minnesota, Twin Cities
[*]Equal contribution

## Abstract

The deployment constraints in practical applications necessitate the pruning of large-scale deep learning models, *i.e.*, promoting their weight sparsity. As illustrated by the Lottery Ticket Hypothesis (LTH), pruning also has the potential of improving their generalization ability. At the core of LTH, iterative magnitude pruning (IMP) is the predominant pruning method to successfully find 'winning tickets'. Yet, the computation cost of IMP grows prohibitively as the targeted pruning ratio increases. To reduce the computation overhead, various efficient 'one-shot' pruning methods have been developed but these schemes are usually unable to find winning tickets as good as IMP. This raises the question of *how to close the gap between pruning accuracy and pruning efficiency?* To tackle it, we pursue the algorithmic advancement of model pruning. Specifically, we formulate the pruning problem from a fresh and novel viewpoint, bi-level optimization (BLO). We show that the BLO interpretation provides a technically-grounded optimization base for an efficient implementation of the pruning-retraining learning paradigm used in IMP. We also show that the proposed bi-level optimization-oriented pruning method (termed BıP) is a special class of BLO problems with a bi-linear problem structure. By leveraging such bi-linearity, we theoretically show that BıP can be solved as easily as first-order optimization, thus inheriting the computation efficiency. Through extensive experiments on *both structured and unstructured pruning* with 5 model architectures and 4 data sets, we demonstrate that BıP can find better winning tickets than IMP in most cases, and is computationally as efficient as the one-shot pruning schemes, demonstrating 2-7$\times$ speedup over IMP for the same level of model accuracy and sparsity. Codes are available at `https://github.com/OPTML-Group/BiP`.

## 1 Introduction

While over-parameterized structures are key to the improved generalization of deep neural networks (DNNs) [1–3], they create new problems – the millions or even billions of parameters not only increase computational costs during inference, but also pose serious deployment challenges on resource-limited devices [4]. As a result, model pruning has seen a lot of research interest in recent years, focusing on reducing model sizes by removing (or pruning) redundant parameters [4–8]. Model sparsity (achieved by pruning) also benefits adversarial robustness [9], out-of-distribution generalization [10], and transfer learning [11]. Some pruning methods (towards structured sparsity) facilitate model deployment on hardware [12, 13].

36th Conference on Neural Information Processing Systems (NeurIPS 2022).

Among various proposed model pruning algorithms [5, 9, 11, 14–27], the heuristics-based Iterative Magnitude Pruning (IMP) is the current dominant approach to achieving model sparsity without suffering performance loss, as suggested and empirically justified by the Lottery Ticket Hypothesis (LTH) [17]. The LTH hypothesizes the existence of a subnetwork (the so-called 'winning ticket') when trained in isolation (*e.g.*, from either a random initialization [17] or an early-rewinding point of dense model training [19]), can match the performance of the original dense model [5, 11, 17–20]. The core idea of IMP is to iteratively prune and retrain the model while progressively pruning a small ratio of the remaining weights in each iteration, and continuing till the desired pruning ratio has been reached. While IMP often finds the winning ticket, it incurs the cost of repeated model retraining from scratch, making it prohibitively expensive for large datasets or large model architectures [28, 29]. To improve the efficiency of model pruning, numerous heuristics-based one-shot pruning methods [17, 21–25], *e.g.*, one-shot magnitude pruning (OMP), have been proposed. These schemes directly prune the model to the target sparsity and are significantly more efficient than IMP. Yet, the promise of current one-shot pruning methods is dataset/model-specific [22, 30] and mostly lies in the low pruning ratio regime [25]. As systematically studied in [22, 31], there exists a clear performance gap between one-shot pruning and IMP. As an alternative to heuristics-based schemes, optimization-based pruning methods [9, 15, 16, 26, 27] still follow the pruning-retraining paradigm and adopt sparsity regularization [32, 33] or parameterized masking [9, 16, 27] to prune models efficiently. However, these methods do not always match the accuracy of IMP and thus have not been widely used to find winning tickets [17, 21, 23–25, 29]. The empirical results showing that optimization underperforms heuristics motivate us to revisit the algorithmic fundamentals of pruning.

To this end, we put forth a novel perspective of model pruning as a bi-level optimization (BLO) problem. In this new formulation, we show that BLO provides a technically-grounded optimization basis for an efficient implementation of the pruning-retraining paradigm, the key algorithmic component used in IMP. To the best of our knowledge, we make the first rigorous connection between model pruning and BLO. Technically, we propose a novel **bi**-level optimization-enabled **p**runing method (termed BIP). We further show how BIP takes advantage of the bi-linearity of the pruning problem to avoid the computational challenges of common BLO methods, and is as efficient as any first-order alternating optimization scheme. Practically, we demonstrate the superiority of the proposed BIP in terms of accuracy, sparsity, and computation efficiency through extensive

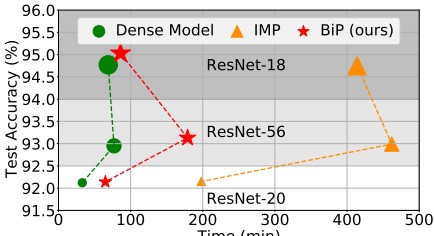

Figure 1: A performance snapshot of the proposed BIP method vs. the IMP baseline and the original dense model across three pruned ResNet models (ResNet-20, ResNet-56, and ResNet-18) with 74% sparsity on CIFAR-10. The marker size indicates the relative model size. The uni-color region corresponds to the same model type used by different pruning methods.

experiments. BIP finds the best winning ticket nearly in all settings while taking time comparable to the one-shot OMP. In Fig. 1, we present a snapshot of our empirical results for CIFAR-10 with 3 ResNet architectures at 74% pruning ratio. In all cases, BIP (★) finds winning tickets, improving accuracy over the dense model (●) and matching IMP (▲), while being upto 5× faster than IMP.

Our **contributions** can be summarized as follows:

● (Formulation) We rethink the algorithmic foundation of model pruning through the lens of BLO (bi-level optimization). The new BLO-oriented formulation disentangles pruning and retraining variables, providing the flexibility to design the interface between pruning and retraining.

● (Algorithm) We propose the new bi-level pruning (BIP) algorithm, which is built upon the aforementioned BLO formulation and the implicit gradient-based optimization theory. Unlike computationally intensive standard BLO solvers, we theoretically show that BIP is as efficient as any first-order optimization by taking advantage of the bi-linear nature of the pruning variables.

● (Experiments) We conduct extensive experiments across 4 datasets (CIFAR-10, CIFAR-100, Tiny-ImageNet and ImageNet), 5 model architectures, and 3 pruning settings (unstructured pruning, filter-wise structured pruning, and channel-wise structured pruning). We show that (i) BIP achieves higher test accuracy than IMP and finds the best winning tickets nearly in all settings, (ii) BIP is highly efficient (comparable to one-shot pruning schemes), that is able to achieve 2-7× speedup over IMP for the same level of model accuracy and sparsity, and (iii) BIP is able to find subnetworks that achieve better performance than the dense model regardless of initialization rewinding.

## 2 Related Work and Open Question

**Neural network pruning.** As neural networks become deeper and more sophisticated, model pruning technology has gained increasing attention over the last decade since pruned models are necessary for the deployment of deep networks in practical applications [4, 34, 35]. With the goal of finding highly-sparse *and* highly-accurate subnetworks from original dense models, a variety of pruning methods have been developed such as heuristics-based pruning [17, 21, 23–25, 29, 36] and optimization-based pruning [9, 16, 26, 27]. The former identifies redundant model weights by leveraging heuristics-based metrics such as weight magnitudes [6, 17, 19, 11, 22, 37, 31, 36, 38], gradient magnitudes [21, 23, 24, 39, 40], and Hessian statistics [41–46].The latter is typically built on: 1) sparsity-promoting optimization [15, 33, 47–50], where model weights are trained by penalizing their sparsity-inducing norms, such as $\ell_0$ and $\ell_1$ norms for irregular weight pruning, and $\ell_2$ norm for structured pruning; 2) parameterized masking [16, 9, 51–55], where model weight scores are optimized to filter the most important weights and achieve better performance.

**Iterative vs. one-shot pruning, and motivation.** Existing schemes can be further categorized into one-shot or iterative pruning based on the pruning schedule employed for achieving the targeted model sparsity. Among the iterative schemes, the IMP (Iterative Magnitude Pruning scheme) [17, 20, 56–65, 36] has played a significant role in identifying high-quality 'winning tickets', as postulated by LTH (Lottery Ticket Hypothesis) [18, 19]. To enable consistent comparisons among different methods, we extend the original definition of winning tickets in [17] to 'matching subnetworks' [20] so as to cover different implementations of winning tickets, *e.g.*, the use of early-epoch rewinding for model re-initialization [18] and the no-rewinding (*i.e.*, fine-tuning) variant [66]. Briefly, the matching subnetworks should match or surpass the performance of the original dense model [20]. In this work, if a matching subnetwork is found better than the winning ticket obtained by the same method that follows the original LTH setup [18, 19], we will also call such a matching subnetwork a winning ticket throughout the paper.

For example, the current state-of-the-art (SOTA) implementation of IMP in [22] can lead to a pruned ResNet-20 on CIFAR-10 with 74% sparsity and 92.12% test accuracy, matching the performance of the original dense model (see red ⋆ in Fig. 2-**Left**). The IMP algorithm typically contains two key ingredients: **(i)** a *temporally-evolving pruning schedule* to progressively increase model sparsity over pruning iterations, and **(ii)** the *pruning-retraining learning mechanism* applied at each pruning iteration. With a target pruning ratio of $p\%$

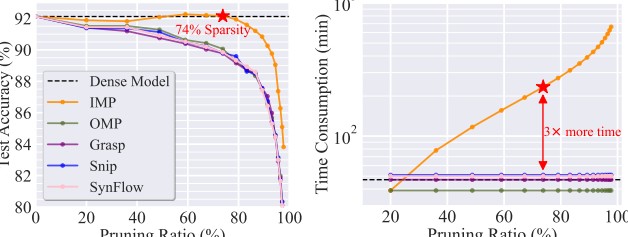

Figure 2: Illustration of the pros and cons of different pruning methods executed over (ResNet-20, CIFAR-10). **Left**: test accuracy vs. different pruning ratios of IMP and one-shot pruning methods (OMP [17], GRASP [23], SNIP [21], SYNFLOW [24]). **Right**: comparison of the efficiency with different sparsity.

with $T$ pruning iterations, an example pruning schedule in (i) could be as follows – each iteration prunes $(p\%)^{1/T}$ of the currently unpruned model weights, progressively pruning fewer weights in each iteration. For (ii), the unpruned weights in each pruning iteration are re-set to the weights at initialization or at an early-training epoch [18], and re-trained till convergence. In brief, IMP repeatedly prunes, resets, and trains the network over multiple iterations.

However, winning tickets found by IMP incur significant computational costs. The sparsest winning ticket found by IMP in Fig. 2-**Left** (red ⋆) utilizes $T = 6$ pruning iterations. As shown in Fig. 2-**Right**, this takes 3× more time than the original training of the dense model. To avoid the computational cost of IMP, different kinds of 'accelerated' pruning methods were developed [17, 21, 23–25, 29], and many fall into the *one-shot pruning* category: The network is directly pruned to the target sparsity and retrained once. In particular, OMP (one-shot magnitude pruning) is an important baseline that simplifies IMP [17]. It follows the pruning-retraining paradigm, but prunes the model to the target ratio with a single pruning iteration. Although one-shot pruning schemes are computationally cheap (Fig. 2-**Right**), they incur a significant accuracy drop compared to IMP (Fig. 2-**Left**). Even if IMP is customized with a reduced number of training epochs per pruning round, the pruning accuracy also drops largely (see Fig. A2). Hence, there is a need for advanced model pruning techniques to find winning tickets like IMP, while being efficient like one-shot pruning.

Different from unstructured weight pruning described above, structured pruning takes into consideration the sparse patterns of the model, such as filter and channel-wise pruning [13, 48, 51, 67–71]. Structured pruning is desirable for deep model deployment in the presence of hardware constraints [4, 34]. However, compared to unstructured pruning, it is usually more challenging to maintain the performance and find structure-aware winning tickets [28, 29].

**Open question.** As discussed above, one-shot pruning methods are unable to match the predictive performance of IMP, and structure-aware winning tickets are hard to find. Clearly, the best optimization foundation of model pruning is underdeveloped. Thus, we ask:

> Is there an **optimization basis** for a successful pruning algorithm that can attain high pruned model accuracy (like IMP) *and* high computational efficiency (like one-shot pruning)?

The model pruning problem has a natural hierarchical structure – we need to find the best mask to prune model parameters, *and then*, given the mask, find the best model weights for the unpruned model parameters. Given this hierarchical structure, we believe that the *bi-level optimization (BLO) framework* is one promising optimization basis for a successful pruning algorithm.

**Bi-level optimization (BLO).** BLO is a general hierarchical optimization framework, where the upper-level problem depends on the solution to the lower-level problem. Such a nested structure makes the BLO in its most generic form very difficult to solve, since it is hard to characterize the influence of the lower-level optimization on the upper-level problem. Various existing literature focuses on the design and analysis of algorithms for various special cases of BLO. Applications range from the classical approximate descent methods [72–74], penalty-based methods [75, 76], to recent BigSAM [77] and its extensions [78, 79]. It is also studied in the area of stochastic algorithms [80–82] and back-propagation based algorithms [83–85]. BLO has also advanced adversarial training [86], meta-learning [87], data poisoning attack generation [88], neural architecture search [89] as well as reinforcement learning [90]. Although BLO was referred in [50] for model pruning, it actually called an ordinary alternating optimization procedure, without taking the hierarchical learning structure of BLO into consideration. To the best of our knowledge, the BLO framework has not been considered for model pruning in-depth and systematically. We will show that model pruning yields a special class of BLO problems with bi-linear optimization variables. We will also theoretically show that this specialized BLO problem for model pruning can be solved as efficiently as first-order optimization. This is in a sharp contrast to existing BLO applications that rely on heuristics-based BLO solvers (*e.g.*, gradient unrolling in meta learning [87] and neural architecture search [89, 91]).

## 3 BIP: Model Pruning via Bi-level Optimization

In this section, we re-investigate the problem of model pruning through the lens of BLO and develop the bi-level pruning (BIP) algorithm. We can theoretically show that BIP can be solved as easily as the first-order alternating optimization by taking advantage of the bi-linearity of pruning variables.

**A BLO viewpoint on model pruning** As described in the previous section, the pruning-retraining learning paradigm covers two kinds of tasks: ❶ pruning that determines the sparse pattern of model weights, and ❷ training remaining non-zero weights to recover the model accuracy. In existing optimization-based pruning methods [92–95], the tasks ❶-❷ are typically achieved by optimizing model weights, together with penalizing their sparsity-inducing norms, *e.g.*, the $\ell_1$ and $\ell_2$ norms [96]. Different from the above formulation, we propose to separate optimization variables involved in the pruning tasks ❶ and ❷. This leads to the (binary) pruning mask variable $\mathbf{m} \in \{0, 1\}^n$ and the model weight variable $\boldsymbol{\theta} \in \mathbb{R}^n$. Here $n$ denotes the total number of model parameters. Accordingly, the pruned model is given by $(\mathbf{m} \odot \boldsymbol{\theta})$, where $\odot$ denotes the element-wise multiplication. As will be evident later, this form of *variable disentanglement* enables us to explicitly depict how the pruning and retraining process co-evolve, and helps customize the pruning task with high flexibility.

We next elaborate on how BLO can be established to co-optimize the pruning mask $\mathbf{m}$ and the retrained sparse model weights $\boldsymbol{\theta}$. Given the pruning ratio $p\%$, the sparsity constraint is given by $\mathbf{m} \in \mathcal{S}$, where $\mathcal{S} = \{\mathbf{m} \mid \mathbf{m} \in \{0, 1\}^n, \mathbf{1}^T \mathbf{m} \leq k\}$ and $k = (1 - p\%)n$. Our goal is to prune the original model *directly* to the targeted pruning ratio $p\%$ (*i.e.*, without calling for the IMP-like sparsity schedule as described in Sec. 2) and obtain the optimized sparse model $(\mathbf{m} \odot \boldsymbol{\theta})$. To this

end, we interpret the pruning task (*i.e.*, ❶) and the model retraining task (*i.e.*, ❷) as *two optimization levels*, where the former is formulated as an *upper-level* optimization problem, and it relies on the optimization of the *lower-level* retraining task. We thus cast the model pruning problem as the following BLO problem (with ❷ being nested inside ❶):

$$
\underbrace{\underset{\mathbf{m}\in\mathcal{S}}{\text{minimize}}\quad \ell(\mathbf{m}\odot\boldsymbol{\theta}^*(\mathbf{m}));}_{\text{❶: Pruning task}} \qquad \text{subject to}\quad \underbrace{\boldsymbol{\theta}^*(\mathbf{m})=\underset{\boldsymbol{\theta}\in\mathbb{R}^n}{\arg\min}\,\overbrace{\ell(\mathbf{m}\odot\boldsymbol{\theta})+\frac{\gamma}{2}\|\boldsymbol{\theta}\|_2^2}^{:=\,g(\mathbf{m},\boldsymbol{\theta})},}_{\text{❷: Sparsity-fixed model retraining}} \tag{1}
$$

where $\ell$ denotes the training loss (*e.g.*, cross-entropy), $\mathbf{m}$ and $\boldsymbol{\theta}$ are the upper-level and lower-level optimization variables respectively, $\boldsymbol{\theta}^*(\mathbf{m})$ signifies the lower-level solution obtained by minimizing the objective function $g$ given the pruning mask $\mathbf{m}$, and $\gamma > 0$ is a regularization parameter introduced to convexify the lower-level optimization so as to stabilize the gradient flow from $\boldsymbol{\theta}^*(\mathbf{m})$ to $\mathbf{m}$ and thus the convergence of BLO [82, 97]. In a sharp contrast to existing single-level optimization-based model pruning methods [92–95], the BLO formulation (1) brings in two advantages.

First, BLO has the flexibility to use mismatched pruning and retraining objectives at the upper and lower optimization levels, respectively. This flexibility allows us to regularize the lower-level training objective function in (1) and customize the implemented optimization methods at both levels. To be more specific, one can update the upper-level pruning mask $\mathbf{m}$ using a data batch (called $\mathcal{B}_2$) distinct from the one (called $\mathcal{B}_1$) used for obtaining the lower-level solution $\boldsymbol{\theta}^*(\mathbf{m})$. The resulting BLO procedure can then mimic the idea of meta learning to improve model generalization [98], where the lower-level problem fine-tunes $\boldsymbol{\theta}$ using $\mathcal{B}_1$, and the upper-level problem validates the generalization of the sparsity-aware finetuned model ($\mathbf{m}\odot\boldsymbol{\theta}^*(\mathbf{m})$) using $\mathcal{B}_2$.

Second, BLO enables us to explicitly model and optimize the coupling between the retrained model weights $\boldsymbol{\theta}^*(\mathbf{m})$ and the pruning mask $\mathbf{m}$ through the implicit gradient (IG)-based optimization routine. Here IG refers to the gradient of the lower-level solution $\boldsymbol{\theta}^*(\mathbf{m})$ with respect to (w.r.t.) the upper-level variable $\mathbf{m}$, and its derivation calls the implicit function theory [76]. The use of IG makes our proposed BLO-oriented pruning (1) significantly different from the greedy alternating minimization [99] that learns the upper-level and lower-level variables independently (*i.e.*, minimizes one variable by fixing the other). We refer readers to the following section for the detailed IG theory. We will also show in Sec. 4 that the pruning strategy from (1) can outperform IMP in many pruning scenarios but is much more efficient as it does not call for the scheduler of iterative pruning ratios.

**Optimization foundation of BIP.** The key optimization challenge of solving the BIP problem (1) lies in the computation of IG (implicit gradient). Prior to developing an effective solution, we first elaborate on the *IG challenge*, the unique characteristic of BLO. In the context of gradient descent, the gradient of the objective function in (1) yields

$$
\underbrace{\frac{d\ell(\mathbf{m}\odot\boldsymbol{\theta}^*(\mathbf{m}))}{d\mathbf{m}}}_{\text{Gradient of objective}}=\nabla_\mathbf{m}\ell(\mathbf{m}\odot\boldsymbol{\theta}^*(\mathbf{m}))+\underbrace{\frac{d(\boldsymbol{\theta}^*(\mathbf{m})^\top)}{d\mathbf{m}}}_{\text{IG}}\nabla_{\boldsymbol{\theta}}\ell(\mathbf{m}\odot\boldsymbol{\theta}^*(\mathbf{m})), \tag{2}
$$

where $\nabla_\mathbf{m}$ and $\nabla_{\boldsymbol{\theta}}$ denote the *partial derivatives* of the bi-variate function $\ell(\mathbf{m}\odot\boldsymbol{\theta})$ w.r.t. the variable $\mathbf{m}$ and $\boldsymbol{\theta}$ respectively, $d\boldsymbol{\theta}^\top/d\mathbf{m}\in\mathbb{R}^{n\times n}$ denotes the vector-wise *full derivative*, and for ease of notation, we will omit the transpose $\top$ when the context is clear. In (2), the IG challenge refers to the demand for computing the full gradient of the implicit function $\boldsymbol{\theta}^*(\mathbf{m})=\arg\min_{\boldsymbol{\theta}}g(\mathbf{m},\boldsymbol{\theta})$ w.r.t. $\mathbf{m}$, where recall from (1) that $g(\mathbf{m},\boldsymbol{\theta}):=\ell(\mathbf{m}\odot\boldsymbol{\theta})+\frac{\gamma}{2}\|\boldsymbol{\theta}\|_2^2$.

Next, we derive the IG formula following the rigorous implicit function theory [76, 82, 87]. Based on the fact that $\boldsymbol{\theta}^*(\mathbf{m})$ satisfies the stationarity condition for the lower-level objective function in (2), it is not difficult to obtain that (see derivation in Appendix A)

$$
\frac{d\boldsymbol{\theta}^*(\mathbf{m})}{d\mathbf{m}}=-\nabla_{\mathbf{m}\boldsymbol{\theta}}^2\ell(\mathbf{m}\odot\boldsymbol{\theta}^*)[\nabla_{\boldsymbol{\theta}}^2\ell(\mathbf{m}\odot\boldsymbol{\theta}^*)+\gamma\mathbf{I}]^{-1}, \tag{3}
$$

where $\nabla_{\mathbf{m}\boldsymbol{\theta}}^2\ell$ and $\nabla_{\boldsymbol{\theta}}^2\ell$ denote the second-order partial derivatives of $\ell$ respectively, and $(\cdot)^{-1}$ denotes the matrix inversion operation.

Yet, the exact IG formula (3) remains difficult to calculate due to the presence of matrix inversion and second-order partial derivatives. To simplify it, we impose the Hessian-free assumption, $\nabla_{\boldsymbol{\theta}}^2\ell=\mathbf{0}$, which is mild in general; For example, the decision boundaries of neural networks with ReLU

activations are piece-wise linear in a tropical hyper-surface [100], and this assumption has been widely used in BLO-involved applications such as meta learning [101] and adversarial learning [86]. Given $\nabla_{\boldsymbol{\theta}}^2 \ell = \mathbf{0}$, the matrix inversion in (3) can be then mitigated, leading to the IG formula

$$\frac{d\boldsymbol{\theta}^*(\mathbf{m})}{d\mathbf{m}} = -\frac{1}{\gamma}\nabla_{\mathbf{m}\boldsymbol{\theta}}^2 \ell(\mathbf{m} \odot \boldsymbol{\theta}^*). \tag{4}$$

At the first glance, the computation of the simplified IG (4) still requires the mixed (second-order) partial derivative $\nabla_{\mathbf{m}\boldsymbol{\theta}}^2 \ell$. However, BIP is a special class of BLO problems with bi-linear variables $(\mathbf{m} \odot \boldsymbol{\theta})$. Based on this bi-linearity, we can prove that IG in (4) can be *analytically* expressed using only *first-order* derivatives; see the following theorem.

**Proposition 1** *Assuming $\nabla_{\boldsymbol{\theta}}^2 \ell = 0$ and defining $\nabla_{\mathbf{z}}\ell(\mathbf{z}) := \nabla_{\mathbf{z}}\ell(\mathbf{z})|_{\mathbf{z}=\mathbf{m} \odot \boldsymbol{\theta}^*}$, the implicit gradient (4) is then given by*

$$\frac{d\boldsymbol{\theta}^*(\mathbf{m})}{d\mathbf{m}} = -\frac{1}{\gamma}\mathrm{diag}(\nabla_{\mathbf{z}}\ell(\mathbf{z})); \tag{5}$$

*Further, the gradient of the objective function given by (2) becomes*

$$\frac{d\ell(\mathbf{m} \odot \boldsymbol{\theta}^*)}{d\mathbf{m}} = (\boldsymbol{\theta}^* - \frac{1}{\gamma}\mathbf{m} \odot \nabla_{\mathbf{z}}\ell(\mathbf{z})) \odot \nabla_{\mathbf{z}}\ell(\mathbf{z}), \tag{6}$$

*where $\odot$ denotes the element-wise multiplication.*

**Proof**: Using chain-rule, we can obtain that

$$\nabla_{\boldsymbol{\theta}}\ell(\mathbf{m} \odot \boldsymbol{\theta}^*) = \mathrm{diag}(\mathbf{m})\nabla_{\mathbf{z}}\ell(\mathbf{z}) = \mathbf{m} \odot \nabla_{\mathbf{z}}\ell(\mathbf{z}); \tag{7}$$

$$\text{similarly,} \quad \nabla_{\mathbf{m}}\ell(\mathbf{m} \odot \boldsymbol{\theta}^*) = \mathrm{diag}(\boldsymbol{\theta}^*)\nabla_{\mathbf{z}}\ell(\mathbf{z}) = \boldsymbol{\theta}^* \odot \nabla_{\mathbf{z}}\ell(\mathbf{z}) \tag{8}$$

where $\mathrm{diag}(\cdot)$ represents a diagonal matrix with $\cdot$ being the main diagonal vector. Further, we can convert (4) to

$$\nabla_{\mathbf{m}\boldsymbol{\theta}}^2 \ell(\mathbf{m} \odot \boldsymbol{\theta}^*) \overset{(7)}{=} \nabla_{\mathbf{m}}\left[\mathbf{m} \odot \nabla_{\mathbf{z}}\ell(\mathbf{z})\right] \overset{\text{chain rule}}{=} \mathrm{diag}(\nabla_{\mathbf{z}}\ell(\mathbf{z})) + \mathrm{diag}(\mathbf{m})[\nabla_{\mathbf{m}}(\nabla_{\mathbf{z}}\ell(\mathbf{z}))]$$

$$\overset{(8)}{=} \mathrm{diag}(\nabla_{\mathbf{z}}\ell(\mathbf{z})) + \mathrm{diag}(\mathbf{m})[\mathrm{diag}(\boldsymbol{\theta}^*)\nabla_{\mathbf{z}}^2\ell(\mathbf{z})] = \mathrm{diag}(\nabla_{\mathbf{z}}\ell(\mathbf{z})), \tag{9}$$

where the last equality holds due to the Hessian-free assumption. With (9) and (4) we can prove (5).

Next, substituting the IG (5) to the upper-level gradient (2), we obtain that

$$\frac{d\ell(\mathbf{m} \odot \boldsymbol{\theta}^*)}{d\mathbf{m}} = \nabla_{\mathbf{m}}\ell(\mathbf{m} \odot \boldsymbol{\theta}^*) - \frac{1}{\gamma}\nabla_{\mathbf{z}}\ell(\mathbf{z}) \odot \nabla_{\boldsymbol{\theta}}\ell(\mathbf{m} \odot \boldsymbol{\theta}^*)$$

$$\overset{(7),(8)}{=} \boldsymbol{\theta}^* \odot \nabla_{\mathbf{z}}\ell(\mathbf{z}) - \frac{1}{\gamma}\nabla_{\mathbf{z}}\ell(\mathbf{z}) \odot (\mathbf{m} \odot \nabla_{\mathbf{z}}\ell(\mathbf{z})) = (\boldsymbol{\theta}^* - \frac{1}{\gamma}\mathbf{m} \odot \nabla_{\mathbf{z}}\ell(\mathbf{z})) \odot \nabla_{\mathbf{z}}\ell(\mathbf{z}),$$

which leads to (6). The proof is now complete. □

The key insight drawn from Prop. 1 is that the bi-linearity of pruning variables (*i.e.*, $\mathbf{m} \odot \boldsymbol{\theta}^*$) makes the IG-involved gradient (2) easily solvable, and the computational complexity is almost the same as that of computing the first-order gradient $\nabla_{\mathbf{z}}\ell(\mathbf{z})$ just once, as supported by (6)

**BIP algorithm and implementation.** We next formalize the BIP algorithm based on Prop. 1 and the alternating gradient descent based BLO solver [82]. At iteration $t$, there are two main steps.

★ *Lower-level SGD for model retraining*: Given $\mathbf{m}^{(t-1)}$, $\boldsymbol{\theta}^{(t-1)}$, and $\mathbf{z}^{(t-1)} := \mathbf{m}^{(t-1)} \odot \boldsymbol{\theta}^{(t-1)}$, we update $\boldsymbol{\theta}^{(t)}$ by randomly selecting a data batch with the learning rate $\alpha$ and applying SGD (stochastic gradient descent) to the lower-level problem of (1),

$$\boldsymbol{\theta}^{(t)} = \boldsymbol{\theta}^{(t-1)} - \alpha\nabla_{\boldsymbol{\theta}}g(\mathbf{m}^{(t-1)}, \boldsymbol{\theta}^{(t-1)}) \overset{(7)}{=} \boldsymbol{\theta}^{(t-1)} - \alpha[\mathbf{m}^{(t-1)} \odot \nabla_{\mathbf{z}}\ell(\mathbf{z})|_{\mathbf{z}=\mathbf{z}^{(t-1)}} + \gamma\boldsymbol{\theta}^{(t-1)}], \quad (\boldsymbol{\theta}\text{-step})$$

★ *Upper-level SPGD for pruning*: Given $\mathbf{m}^{(t-1)}$, $\boldsymbol{\theta}^{(t)}$, and $\mathbf{z}^{(t+1/2)} := \mathbf{m}^{(t-1)} \odot \boldsymbol{\theta}^{(t)}$, we update $\mathbf{m}$ using SPGD (stochastic projected gradient descent) along the IG-enhanced descent direction (2),

$$\mathbf{m}^{(t)} = \mathcal{P}_{\mathcal{S}}\left[\mathbf{m}^{(t-1)} - \beta\frac{d\ell(\mathbf{m} \odot \boldsymbol{\theta}^{(t)})}{d\mathbf{m}}|_{\mathbf{m}=\mathbf{m}^{(t-1)}}\right]$$

$$\overset{(6)}{=} \mathcal{P}_{\mathcal{S}}\left[\mathbf{m}^{(t-1)} - \beta\left(\boldsymbol{\theta}^{(t)} - \frac{1}{\gamma}\mathbf{m}^{(t-1)} \odot \nabla_{\mathbf{z}}\ell(\mathbf{z})|_{\mathbf{z}=\mathbf{z}^{(t+1/2)}}\right) \odot \nabla_{\mathbf{z}}\ell(\mathbf{z})|_{\mathbf{z}=\mathbf{z}^{(t+1/2)}}\right], \quad (\mathbf{m}\text{-step})$$

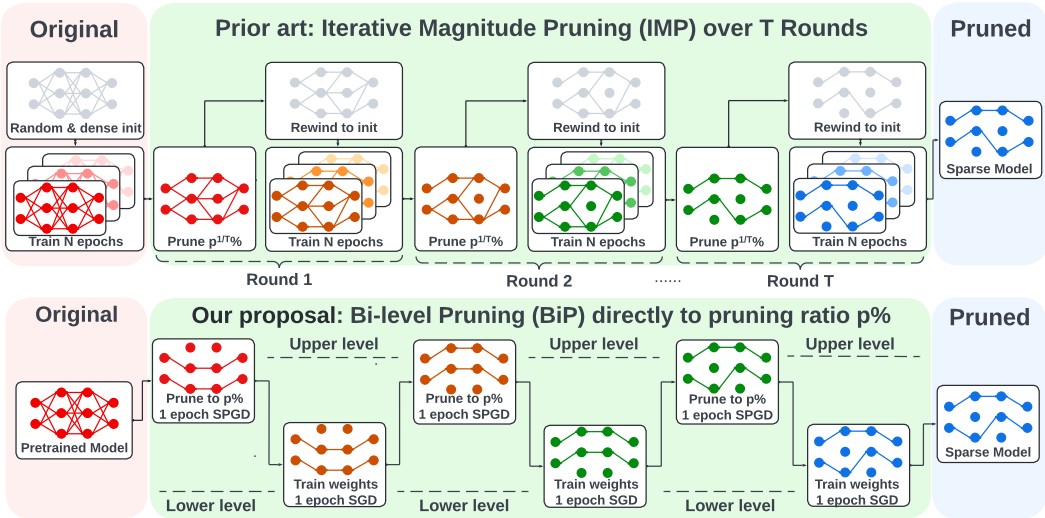

Figure 3: Visualization of pruning pipeline comparison between IMP and BιP. Edge refers to the mask update and color refers to the weight update.

where $\beta > 0$ is the upper-level learning rate, and $\mathcal{P}_\mathcal{S}(\cdot)$ denotes the Euclidean projection onto the constraint set $\mathcal{S}$ given by $\mathcal{S} = \{\mathbf{m} \mid \mathbf{m} \in \{0,1\}^n, \mathbf{1}^T\mathbf{m} \leq k\}$ in (1) and is achieved by the top-$k$ hard-thresholding operation as will be detailed later.

In BιP, the ($\boldsymbol{\theta}$-step) and (m-step) steps execute iteratively. For clarity, Fig. 3 shows the difference between the pruning pipelines of BιP and IMP. In contrast to IMP that progressively prunes and retrains a model with a growing pruning ratio, BιP directly prunes the model to the targeted sparsity level without involving costly re-training process. In practice, we find that both the upper- and lower-level optimization routines of BιP converge very well (see Fig. A12 and Fig. A13). It is also worth noting that both ($\boldsymbol{\theta}$-step) and (m-step) only require the first-order information $\nabla_\mathbf{z}\ell(\mathbf{z})$, demonstrating that BιP can be conducted as efficiently as first-order optimization. In Fig. A1, we highlight the algorithmic details on the BιP pipeline. We present more implementation details of BιP below and refer readers to Appendix B for a detailed algorithm description.

✦ *Discrete optimization over* $\mathbf{m}$: We follow the 'convex relaxation + hard thresholding' mechanism used in [9, 16]. Specifically, we relax the binary masking variables to continuous masking scores $\mathbf{m} \in [\mathbf{0}, \mathbf{1}]$. We then acquire loss gradients at the backward pass based on the relaxed $\mathbf{m}$. At the forward pass, we project it onto the discrete constraint set $\mathcal{S}$ using the hard thresholding operator, where the top $k$ elements are set to 1s and the others to 0s. See Appendix B for more discussion.

✦ *Data batch selection for lower-level and upper-level optimization:* We adopt different data batches (with the same batch size) when implementing ($\boldsymbol{\theta}$-step) and (m-step). This is one of the advantages of the BLO formulation, which enables the flexibility to customize the lower-level and upper-level problems. The use of diverse data batches is beneficial to generalization as shown in [98].

✦ *Hyperparameter tuning:* As described in ($\boldsymbol{\theta}$-step)-(m-step), BιP needs to set two learning rates $\alpha$ and $\beta$ for lower-level and upper-level optimization, respectively. We choose $\alpha = 0.01$ and $\beta = 0.1$ in all experiments, where we adopt the mask learning rate $\beta$ from Hydra [9] and set a smaller lower-level learning rate $\alpha$, as $\boldsymbol{\theta}$ is initialized by a pre-trained dense model. We show ablation study on $\alpha$ in Fig. A8(c). BLO also brings in the low-level convexification parameter $\gamma$. We set $\gamma = 1.0$ in experiments and refer readers to Fig. A8(b) for a sanity check.

✦ *One-step vs. multi-step SGD:* In ($\boldsymbol{\theta}$-step), the one-step SGD is used and helps reduce the computation overhead. In practice, we also find that the one-step SGD is sufficient: The use of multi-step SGD in BιP does not yield much significant improvement over the one-step version; see Fig. A8(a).

✦ *Extension to structured pruning:* We formulate and solve the BιP problem in the context of unstructured (element-wise) weight pruning. However, if define the pruning mask $\mathbf{m}$ *w.r.t.* model's structural units (*e.g.*, filters), BιP is easily applied to structured pruning (see Fig. 6 and Fig. A10).

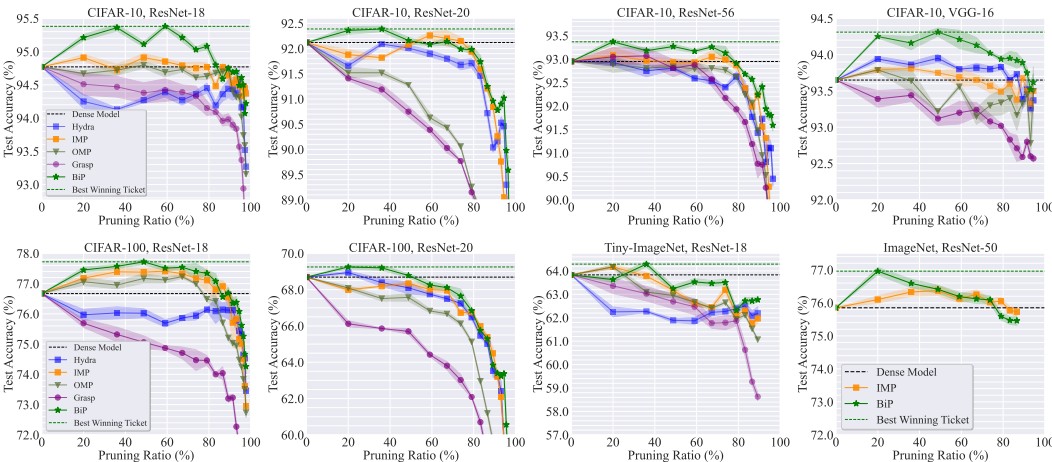

Figure 4: Unstructured pruning trajectory given by test accuracy (%) vs. sparsity (%) on various dataset-model pairs. The proposed BIP is compared with HYDRA [9], IMP [22], OMP [22], GRASP [23]. And the performance of dense model and that of the best winning ticket are marked using dashed lines in each plot. The solid line and shaded area of each pruning method represent the mean and variance of test accuracies over 3 independent trials. We observe that BIP consistently outperforms the other baselines. Note in the (ImageNet, ResNet-50) setting, we only compare BIP with our strongest baseline IMP due to computational resource constraints.

## 4 Experiments

In this section, we present extensive experimental results to show the effectiveness of BIP across multiple model architectures, various datasets, and different pruning setups. Compared to IMP, one-shot pruning, and optimization-based pruning baselines, we find that BIP can find better winning tickets in most cases and is computationally efficient.

### 4.1 Experiment Setup

**Datasets and models.** Following the pruning benchmark in [22], we consider 4 datasets including CIFAR-10 [102], CIFAR-100 [102], Tiny-ImageNet [103], ImageNet [104], and 5 architecture types including ResNet-20/56/18/50 and VGG-16 [105, 106]. Tab. A1 summarizes these datasets and model configurations and setups.

**Baselines, training, and evaluation.** As baselines, we mainly focus on 4 SOTA pruning methods, ① IMP [17], ② OMP [17], ③ GRASP [23] (a one-shot pruning method by analyzing gradient flow at initialization), and ④ HYDRA [9] (an optimization-based pruning method that optimizes masking scores). It is worth noting that there exist various implementations of IMP, *e.g.*, specified by different learning rates and model initialization or 'rewinding' strategies [18]. To make a fair comparison, we follow the recent IMP benchmark in [22], which can find the best winning tickets over current heuristics-based pruning baselines. We also remark that HYDRA is originally proposed for improving the adversarial robustness of a pruned model, but it can be easily customized for standard pruning when setting the adversary's strength as 0 [9]. We choose HYDRA as a baseline because it can be regarded as a single-level variant of BIP with post-optimization weight retraining. When implementing BIP, unless specified otherwise, we use the 1-step SGD in ($\theta$-step), and set the learning rates $(\alpha, \beta)$ and the lower-level regularization parameter $\gamma$ as described in the previous section. When implementing baselines, we follow their official repository setups. We evaluate the performance of all methods mainly from two perspectives: (1) the test accuracy of the sub-network, and (2) the runtime of pruning to reach the desired sparsity. We refer readers to Tab. A3 and Appendix C.2 for more training and evaluation details, such as training epochs and learning rate schedules.

### 4.2 Experiment Results

**BIP identifies high-accuracy subnetworks.** In what follows, we look at the quality of winning tickets identified by BIP. *Two key observations* can be drawn from our results: (1) BIP finds winning tickets of higher accuracy and/or higher sparsity than the baselines in most cases (as shown in Fig. 4

Table 1: The sparsest winning tickets found by different methods at different data-model setups. Winning tickets refer to the sparse models with an average test accuracy no less than the dense model [20]. In each cell, $p\%$ (acc±std%) represents the sparsity as well as the test accuracy. The test accuracy of dense models can be found in the header. ✗ signifies that no winning ticket is found by a pruning method. Given the data-model setup (*i.e.*, per column), the sparsest winning ticket is highlighted in **bold**.

| Method | CIFAR-10 | | | | CIFAR-100 | |
| --- | --- | --- | --- | --- | --- | --- |
| | ResNet-18 (94.77%) | ResNet-20 (92.12%) | ResNet-56 (92.95%) | VGG-16 (93.65%) | ResNet-18 (76.67%) | ResNet-20 (68.69%) |
| IMP | 87% (94.77±0.10%) | **74%** (92.15±0.15%) | **74%** (92.99±0.12%) | 89% (93.68±0.05%) | 87% (76.91±0.19%) | ✗ |
| OMP | 49% (94.80±0.10%) | ✗ | ✗ | 20% (93.79±0.06%) | 74% (76.99±0.07%) | ✗ |
| GRASP | ✗ | ✗ | 36% (93.07±0.34%) | ✗ | ✗ | ✗ |
| HYDRA | ✗ | ✗ | ✗ | 87% (93.73±0.03%) | ✗ | 20% (68.94±0.17%) |
| BIP | **89%** (94.79±0.15%) | 67% (92.14±0.15%) | **74%** (93.13±0.04%) | **93%** (93.75±0.15%) | **89%** (76.69±0.18%) | **49%** (68.78±0.10%) |

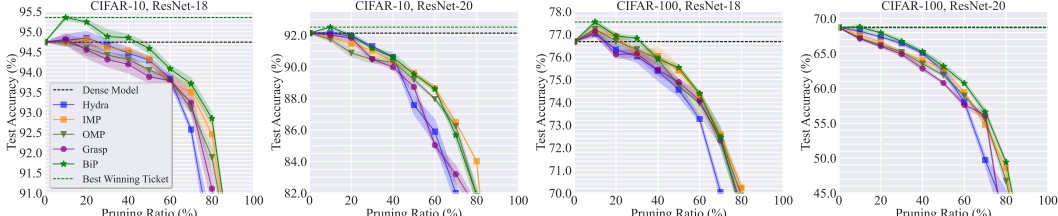

Figure 6: Filter pruning given by test accuracy (%) vs. pruning ratio (%). The visual presentation setting is the same as Fig. 4. We observe that BIP identifies winning tickets of structured pruning in certain sparsity regimes.

and Tab. 1); (2) The superiority of BIP holds for both unstructured pruning and structured pruning (as shown in Fig. 6 and Fig. A10). We refer to more experiment results in Appendix C.3.

Fig. 4 shows the *unstructured pruning trajectory* (given by test accuracy vs. pruning ratio) of BIP and baseline methods in 8 model-dataset setups. For comparison, we also present the performance of the original dense model. As we can see, the proposed BIP approach finds the best winning tickets (in terms of the highest accuracy) compared to the baselines across all the pruning setups. Among the baseline methods, IMP is the most competitive method to ours. However, the improvement brought by BIP is significant with respect to the variance of IMP, except for the 60%-80% sparsity regime in (CIFAR-10, ResNet-20). In the case of (CIFAR-100, ResNet-20), where IMP can not find any winning tickets (as confirmed by [22]), BIP still manages to find winning tickets with around 0.6% improvement over the dense model. In Tab. 1, we summarize the sparsest winning tickets along the pruning trajectory identified by different pruning methods. BIP can identify the winning tickets with higher sparsity levels than the other methods, except in the case of (CIFAR-10, ResNet-20).

Fig. 6 demonstrates the *structured pruning trajectory* on the CIFAR-10/100 datasets. Here we focus on filter pruning, where the filter is regarded as a masking unit in (1). We refer readers to Fig. A10 for channel-wise pruning results. Due to the page limit, we only report the results of the filter-wise pruning in the main paper and please refer to Appendix C.3 for channel-wise pruning. Compared to Fig. 4, Fig. 6 shows that it becomes more difficult to find winning tickets of high accuracy and sparsity in the structured pruning, and the gap among different methods decreases. This is not surprising, since filter pruning imposes much stricter pruning structure constraints than irregular pruning. However, BIP still outperforms all the baselines. Most importantly, it identifies clear winning tickets in the low sparse regime even when IMP fails.

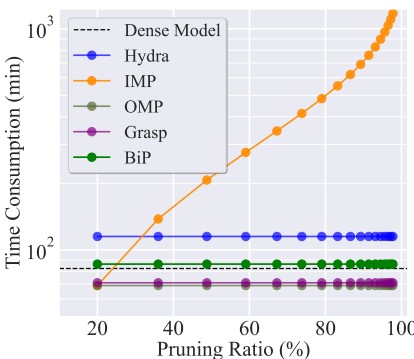

Figure 5: Time consumption comparison on (CIFAR-10, ResNet-18) with different pruning ratio $p$.

**BIP is computationally efficient.** In our experiments, another **key observation** is that BIP yields sparsity-agnostic runtime complexity while IMP leads to runtime exponential to the target sparsity. Fig. 5 shows the computation cost of different methods versus pruning ratios on (CIFAR-10, ResNet-18). For example, BIP takes 86 mins to find the sparsest winning ticket (with 89% sparsity in Tab. 1). This yields 7× less runtime than IMP, which consumes 620 mins to find a comparable winning ticket with 87% sparsity. Compared to the optimization-based baseline HYDRA, BIP is more efficient as it

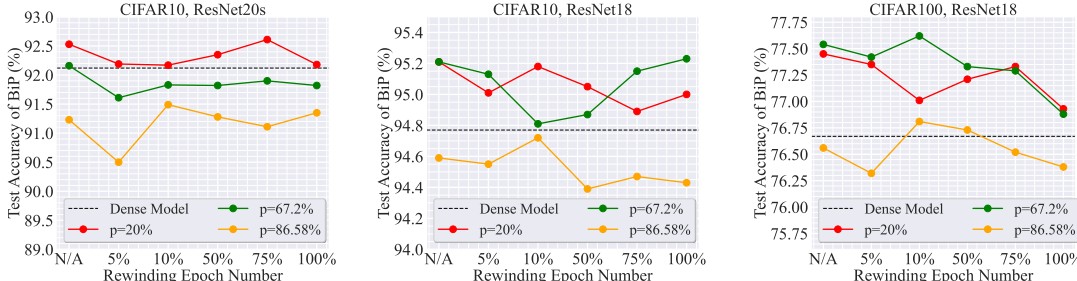

Figure 7: The sensitivity of BIP to rewinding epoch numbers on different datasets and model architectures. "N/A" in the x-axis indicates BIP without retraining.

does not rely on the extra post-optimization retraining; see Tab. A2 for a detailed summary of runtime and number of training epochs required by different pruning methods. Further, BIP takes about 1.25 × more computation time than GRASP and OMP. However, the latter methods lead to worse pruned model accuracy, as demonstrated by their failure to find winning tickets in Tab. 1, Fig. 4, and Fig. 6.

**BIP requires no rewinding.** Another advantage of BIP is that it insensitive to model rewinding to find matching subnetworks. Recall that rewinding is a strategy used in LTH [19] to determine what model initialization should be used for retraining a pruned model. As shown in [22], an IMP-identified winning ticket could be sensitive to the choice of a rewinding point. Fig. 7 shows the test accuracy of the BIP-pruned model when it is retrained at different rewinding epochs under various datasets and model architectures, where 'N/A' in the x-axis represents the case of no retraining (and thus no rewinding). As we can see, a carefully-tuned rewinding scheme does not lead to a significant improvement over BIP without retraining. This suggests that the subnetworks found by BIP are already of high quality and does not require any rewinding operation.

**Additional results.** We include more experiment results in Appendix C.3. In particular, we show more results in both unstructured and structured pruning settings in Fig. A4, Fig. A5, Fig. A6 and Fig. A7, where we compare BIP with more baselines and cover more model architectures. We also study the sensitivity of BIP to the lower-level step number, lower-level regularization coefficient, the significance of the implicit gradient term (2), learning rate, and batch size, as shown in Fig. A8 and Fig. A9. To demonstrate the convergence of the upper-level and lower-level optimization in BIP, we show the training trajectory of BIP for accuracy (Fig.A12) and mask score (Fig.A13), and show how the lower-level step number affects the convergence speed (Fig. A14)). Further, we show the performance of BIP vs. the growth of training epochs (Fig. A15), and its performance vs. different data batch schedulers (see Fig. A16).

## 5   Conclusion

We proposed the BIP method to find sparse networks through the lens of BLO. Our work advanced the algorithmic foundation of model pruning by characterizing its pruning-retraining hierarchy using BLO. We theoretically showed that BIP can be solved as easily as first-order optimization by exploiting the bi-linearity of pruning variables. We also empirically showed that BIP can find high-quality winning tickets very efficiently compared to the predominant iterative pruning method. In the future, we will seek the optimal curriculum of training data at different optimization levels of BIP, and will investigate the performance of our proposal for actual hardware acceleration.

## Acknowledgement

The work of Y. Zhang, Y. Yao, and S. Liu was supported by National Science Foundation (NSF) Grant IIS-2207052. The work of M. Hong was supported by NSF grants CIF-1910385 and CMMI-1727757. The work of Y. Wang was supported NSF grant CCF-1919117. The computing resources used in this work were also supported by the MIT-IBM Watson AI Lab, IBM Research and the Institute for Cyber-Enabled Research (ICER) at Michigan State University.

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
