# Appendix

## A Derivation of (3)

Based on the fact that the $\boldsymbol{\theta}^*(\mathbf{m})$ is satisfied with the stationary condition of the lower-level objective function in (2), we obtain

$$\nabla_{\boldsymbol{\theta}} g(\mathbf{m}, \boldsymbol{\theta}^*) = \nabla_{\boldsymbol{\theta}} \ell(\mathbf{m} \odot \boldsymbol{\theta}^*) + \gamma \boldsymbol{\theta}^* = \mathbf{0}, \tag{A1}$$

where for ease of notation, we omit the dependence of $\boldsymbol{\theta}^*(\mathbf{m})$ w.r.t. $\mathbf{m}$. We then take derivative of the second equality of (A1) w.r.t. $\mathbf{m}$ by using the implicit function theory. This leads to

$$\nabla^2_{\mathbf{m}\boldsymbol{\theta}} \ell(\mathbf{m} \odot \boldsymbol{\theta}^*) + \frac{d\boldsymbol{\theta}^*(\mathbf{m})}{d\mathbf{m}} \nabla^2_{\boldsymbol{\theta}} \ell(\mathbf{m} \odot \boldsymbol{\theta}^*) + \gamma \frac{d\boldsymbol{\theta}^*(\mathbf{m})}{d\mathbf{m}} = \mathbf{0};$$

$$\implies \frac{d\boldsymbol{\theta}^*(\mathbf{m})}{d\mathbf{m}} = -\nabla^2_{\mathbf{m}\boldsymbol{\theta}} \ell(\mathbf{m} \odot \boldsymbol{\theta}^*) [\nabla^2_{\boldsymbol{\theta}} \ell(\mathbf{m} \odot \boldsymbol{\theta}^*) + \gamma \mathbf{I}]^{-1}. \tag{A2}$$

## B BIP Algorithm Details

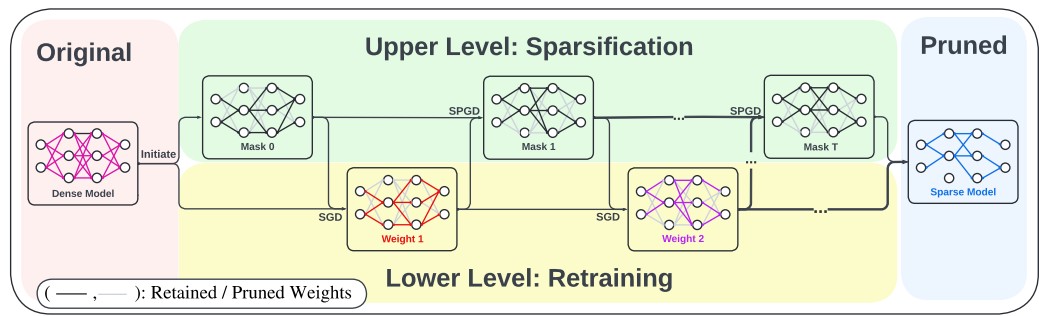

Figure A1: Overview of the BIP pruning algorithm. The BIP algorithm iteratively carry out model retraining in the lower level and pruning in the upper level. In the plots, SGD refers to the lower-level stochastic gradient descent update and SPGD refers to the upper-level stochastic projected gradient descent. Masks may vary between each iteration, and the pruned weights are indicated using the light gray color. Different colors of the edges in the neural networks refer to the weight update. The arcs in this figure represent the data flow of weights/connections.

At iteration $t$ of BIP, there are two main steps:

★ *Lower-level SGD for model retraining*: Given $\mathbf{m}^{(t-1)}$, $\boldsymbol{\theta}^{(t-1)}$, and $\mathbf{z}^{(t-1)} := \mathbf{m}^{(t-1)} \odot \boldsymbol{\theta}^{(t-1)}$, we update $\boldsymbol{\theta}^{(t)}$ by applying SGD (stochastic gradient descent) to the lower-level problem of (1),

$$\boldsymbol{\theta}^{(t)} = \boldsymbol{\theta}^{(t-1)} - \alpha \nabla_{\boldsymbol{\theta}} g(\mathbf{m}^{(t-1)}, \boldsymbol{\theta}^{(t-1)}) \overset{(7)}{=} \boldsymbol{\theta}^{(t-1)} - \alpha[\mathbf{m}^{(t-1)} \odot \nabla_{\mathbf{z}} \ell(\mathbf{z})|_{\mathbf{z}=\mathbf{z}^{(t-1)}} + \gamma \boldsymbol{\theta}^{(t-1)}], \quad (\boldsymbol{\theta}\text{-step})$$

where $\alpha > 0$ is the lower-level learning rate.

★ *Upper-level SPGD for pruning*: Given $\mathbf{m}^{(t-1)}$, $\boldsymbol{\theta}^{(t)}$, and $\mathbf{z}^{(t+1/2)} := \mathbf{m}^{(t-1)} \odot \boldsymbol{\theta}^{(t)}$, we update $\mathbf{m}$ using PGD (projected gradient descent) along the IG-enhanced descent direction (2),

$$\mathbf{m}^{(t)} = \mathcal{P}_{\mathcal{S}}\left[\mathbf{m}^{(t-1)} - \beta \frac{d\ell(\mathbf{m} \odot \boldsymbol{\theta}^{(t)})}{d\mathbf{m}}\Big|_{\mathbf{m}=\mathbf{m}^{(t-1)}}\right]$$

$$\overset{(6)}{=} \mathcal{P}_{\mathcal{S}}\left[\mathbf{m}^{(t-1)} - \beta \left(\boldsymbol{\theta}^{(t)} - \frac{1}{\gamma}\mathbf{m}^{(t-1)} \odot \nabla_{\mathbf{z}} \ell(\mathbf{z})|_{\mathbf{z}=\mathbf{z}^{(t+1/2)}}\right) \odot \nabla_{\mathbf{z}} \ell(\mathbf{z})|_{\mathbf{z}=\mathbf{z}^{(t+1/2)}}\right], \quad (\mathbf{m}\text{-step})$$

where $\beta > 0$ is the upper-level learning rate, and $\mathcal{P}_{\mathcal{S}}(\cdot)$ denotes the Euclidean projection onto the constraint set $\mathcal{S}$ given by $\mathcal{S} = \{\mathbf{m} \,|\, \mathbf{m} \in \{0,1\}^n, \mathbf{1}^T\mathbf{m} \leq k\}$ in (1) and is achieved by the top-$k$ hard-thresholding operation as will be detailed below.

**Implementation of discrete optimization.** In the actual implementation, we use $\widetilde{\mathbf{m}}^{(t)} \in [0,1]^d$ and obtain $\mathbf{m}^{(t)} \in \{0,1\}^d$ where $\mathbf{m}^{(t)} \leftarrow \mathcal{P}_{\mathcal{S}}[\widetilde{\mathbf{m}}^{(t)}]$. The ($\mathbf{m}$-step) is then implemented as the following:

Table A1: Dataset and model setups. The following parameters are shared across all the methods.

| Settings | CIFAR-10 | | | | CIFAR-100 | | Tiny-ImageNet | ImageNet |
|---|---|---|---|---|---|---|---|---|
| | RN-18 | RN-20 | RN-56 | VGG-16 | RN-18 | RN-20 | RN-18 | RN-50 |
| Batch Size | 64 | 64 | 64 | 64 | 64 | 64 | 32 | 1024 |
| Model Size | 11.22 M | 0.27 M | 0.85 M | 14.72 M | 11.22 M | 0.27 M | 11.22 M | 25.56 M |

$$\widetilde{\mathbf{m}}^{(t)} = \widetilde{\mathbf{m}}^{(t-1)} - \beta \left( \boldsymbol{\theta}^{(t)} - \frac{1}{\gamma} \widetilde{\mathbf{m}}^{(t-1)} \odot \nabla_{\mathbf{z}} \ell(\mathbf{z}) \big|_{\mathbf{z} = \mathbf{z}^{(t+1/2)}} \right) \odot \nabla_{\mathbf{z}} \ell(\mathbf{z}) \big|_{\mathbf{z} = \mathbf{z}^{(t+1/2)}} \qquad (\widetilde{\mathbf{m}}\text{-step})$$

and then $\mathbf{m}^{(t)} \leftarrow \mathcal{P}_{\mathcal{S}}\left[ \widetilde{\mathbf{m}}^{(t)} \right]$, with $\mathbf{z}^{(t+1/2)} := \mathbf{m}^{(t-1)} \odot \boldsymbol{\theta}^{(t)}$ as defined above.

---

**Algorithm A1** BIP

1: **Initialize:** Model $\boldsymbol{\theta}_0$, pruning mask score $\mathbf{m}_0$, binary mask $\mathbf{z}^*$, sparse ratio $p\%$, regularization parameter $\lambda$, upper- and lower-level learning rate $\alpha$ and $\beta$.
2: **for** Iteration $t = 0, 1, \ldots,$ **do**
3:     Pick *different* random data batches $\mathcal{B}_\alpha$ and $\mathcal{B}_\beta$ for different levels of tasks.
4:     **Lower-level**: Update model parameters using data batch $\mathcal{B}_\beta$ via SGD calling:

$$\boldsymbol{\theta}_{t+1} = \boldsymbol{\theta}_t - \beta \frac{d\ell_{\mathrm{tr}}(\mathbf{m} \odot \boldsymbol{\theta})}{d\boldsymbol{\theta}} \bigg|_{\mathbf{m} = \mathbf{z}^*, \boldsymbol{\theta} = \boldsymbol{\theta}_t} \qquad (A3)$$

5:     **Upper-level**: Update pruning mask score using data batch $\mathcal{B}_\alpha$ via SGD calling:

$$\mathbf{m}_{t+1} = \mathbf{m}_t - \alpha \left( \nabla_{\mathbf{m}} \ell_{\mathrm{tr}}(\mathbf{m} \odot \boldsymbol{\theta}) - \frac{1}{\gamma} \nabla_{\mathbf{z}} \ell_{\mathrm{tr}}(\mathbf{z}) |_{\mathbf{z} = \mathbf{m} \odot \boldsymbol{\theta}} \odot \nabla_{\boldsymbol{\theta}} \ell_{\mathrm{tr}}(\mathbf{m} \odot \boldsymbol{\theta}) \right) \bigg|_{\mathbf{m} = \mathbf{m}_t, \boldsymbol{\theta} = \boldsymbol{\theta}_{t+1}} \qquad (A4)$$

6:     **Update the binary mask $\mathbf{z}^*$**: Hard-threshold the mask score $\mathbf{m}$ with the give sparse ratio $p$:

$$\mathbf{z}^* = \mathcal{T}_{\{0,1\}^d}(\mathbf{m}_{t+1}, s). \qquad (A5)$$

7: **end for**

---

## C   Additional Experimental Details and Results

### C.1   Datasets and Models

Our dataset and model choices follow the pruning benchmark in [22]. We summarize the datasets and model configurations in Tab. A1. In particular, we would like to stress that we adopt the ResNet-18 with convolutional kernels of $3 \times 3$ in the first layer for Tiny-ImageNet, aligned with CIFAR-10 and CIFAR-100, compared to ImageNet ($7 \times 7$). See `https://github.com/kuangliu/pytorch-cifar/blob/master/models/resnet.py` for more details.

### C.2   Detailed Training Settings

**Baselines.** For both unstructured and structured pruning settings, we consider four baseline methods across various pruning categories, including IMP [17], OMP [17], HYDRA [9] and GRASP [23]. For HYDRA and GRASP, we adopt the original setting as well as hyper-parameter choices on their official code repositories. For IMP and OMP, we adopt the settings from the current SOTA implementations [22]. Details on the pruning schedules can be found in Tab. A2. In particular, HYDRA prunes the dense model to the desired sparsity with 100 epoch for pruning and 100 epoch for retraining. GRASP conducts one-shot pruning to the target sparsity, followed by the 200-epoch retraining. In each pruning iteration, IMP prunes 20% of the remaining parameters before 160-epoch retraining. HYDRA adopts the cosine learning rate scheduler for pruning and retraining stage. The learning rate scheduler for IMP, OMP, and GRASP is the step learning rate with a learning rate decay rate of 0.1 at 50% and 75% epochs. The initial learning rate for all the methods are 0.1.

Table A2: Computation complexities of different pruning methods on (CIFAR-10, ResNet-18) in unstructured pruning setting. The training epoch numbers of pruning/retraining baselines are consistent with their official settings or the latest benchmark implementations. All the evaluations are based on a single Tesla-V100 GPU.

| Method \ Sparsity | Runtime v.s. targeted sparsity | | | | Training epoch # |
| | 20% | 59% | 83.2% | 95.6% | |
|---|---|---|---|---|---|
| IMP | 69 min | 276 min | 621 min | 966 min | 160 epoch retrain |
| GRASP | | 89 min | | | 200 epoch retrain |
| OMP | | 69 min | | | 160 epoch retrain |
| HYDRA | | 115 min | | | 100 epoch prune 100 epoch retrain |
| BIP | | 86 min | | | 100 epoch |

Table A3: Detailed training details for each method. All the baselines adopt the recommended settings either from the official or their latest benchmark (*e.g.*, LTH[22]) for a fair comparison. Note by default setting, only our method BIP do not require additional epochs for retraining.

| Method | Epoch Number | Initial Learning Rate | Learning Rate Scheduler | Learning Rate Decay Factor | Learning Rate Decay Epoch | Mementum | Weight Decay | Rewind Epoch | Warm-up |
|---|---|---|---|---|---|---|---|---|---|
| IMP | 160 for Retrain | 0.1 | Step LR | 10 | 80/120 | 0.9 | 5.00E-04 | 8 | 75 for VGG16 |
| OMP | 160 for Retrain | 0.1 | Step LR | 10 | 80/120 | 0.9 | 5.00E-04 | 8 | 75 for VGG16 |
| HYDRA | 100 for Prune 100 for Retrain | 0.1 | Cosine LR | N/A | N/A | 0.9 | 5.00E-04 | N/A | 75 for VGG16 |
| GRASP | 200 for Retrain | 0.1 | Step LR | 10 | 100/150 | 0.9 | 5.00E-04 | N/A | 75 for VGG16 |
| BIP | 100 | 0.1 for $\mathbf{m}$; 0.01 for $\boldsymbol{\theta}$ | Cosine LR | N/A | N/A | 0.9 | 5.00E-04 | N/A | 75 for VGG16 |

**Hyper-parameters for BIP.** In both structured and unstructured pruning settings, cosine learning rate schedulers are adopted, and BIP takes an initial learning rate of 0.1 for the upper-level problem (pruning) and 0.01 for the lower-level problem (retraining). The lower-level regularization coefficient $\lambda$ is set to 1.0 throughout the experiments. By default, we only take one SGD step for lower-level optimization in all settings. Ablation studies on different SGD steps for lower-level optimization can be found in Fig. A8(b) and Fig. A14. We use 100 training epochs for BIP, and ablation studies on different training epochs for larger pruning ratios can be found in Fig. A15.

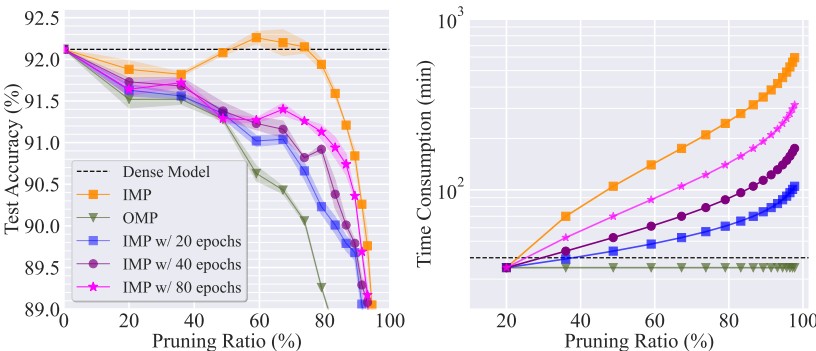

Figure A2: Performance comparisons among OMP, IMP, and IMP with less retraining epochs on CIFAR-10 with ResNet-20.

**Structured pruning.** To differentiate of the filter-wise and channel-wise structured pruning setting, we illustrate the details of these settings in Fig. A3. Note, the filter-wise pruning setting prunes the output dimension (output channel) of the parameters in one layer, while the channel-wise prunes the input dimension (input channel).

### C.3   Additional Experiment Results

**Comparison with IMP using reduced retraining epochs.** As IMP is significantly more time-consuming than one-shot pruning methods, a natural way to improve the efficiency is to decrease the retraining epoch numbers at each pruning cycle. In Fig. A2, the performance and time consumption of

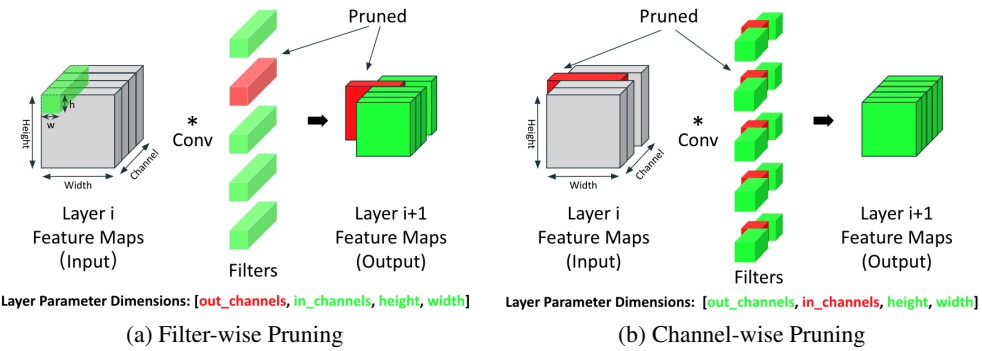

Layer Parameter Dimensions: [out_channels, in_channels, height, width]

(a) Filter-wise Pruning

Layer Parameter Dimensions: [out_channels, in_channels, height, width]

(b) Channel-wise Pruning

Figure A3: Illustration of filter-wise pruning and channel-wise pruning. The blocks in the middle column in (a) and (b) represent the parameters (filters) of the $i$th convolutional layer, where the red ones represent the pruning unit in each setting. The left blocks in gray denote the input feature maps and the right columns denote the output feature maps generated by the corresponding filters marked in the same color.

IMP using 20, 40, and 80 epochs at each retraining cycle are presented. The results and conclusions are in general aligned with Fig. 2. First, with fewer epoch numbers, the time consumption decreases at the cost of evident performance degradation. Second, IMP with fewer epoch numbers are unable to obtain winning tickets. Thus, the direct simplification of IMP would hamper the pruning accuracy. This experiment shows the difficulty of achieving efficient and effective pruning under the scope of heuristics-based pruning, and thus justifies the necessity in developing a more powerful optimization-based pruning method.

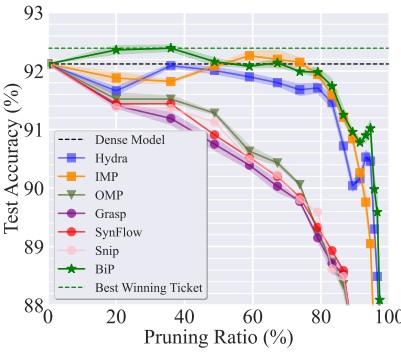

Figure A4: Unstructured pruning performance of BiP vs. prune-at-initialization baselines on (CIFAR-10, ResNet-20).

**Comparison with more prune-at-initialization baselines.**    In Fig. A4, we include more heuristics-based one-shot pruning baselines (SYNFLOW [24], SNIP [21]) for comparison. Together with GRASP, these methods belong to the category of pruning at initialization, which determines the sparse sub-networks prior to training. As we can see, the advantage of our method over the newly added methods are clear, and the benefit becomes more significant as the sparsity increases. This further demonstrates the superiority of the optimization-basis of BiP over the heuristics-based one-shot methods.

**Experiments on unstructured pruning with more baselines.**    We compare our proposed method BiP to more baselines on different datasets and architectures in Fig. A5. We add two more baselines, including EARLYBIRD [28] and PROSPR [25]. The results show that PROSPR is indeed better than GRASP but is still not as good as IMP and our method BiP in different architecture-dataset combinations. Meanwhile, except for the unstructured pruning settings of ResNet18 pruning over CIFAR10 and CIFAR100, PROSPR, as a pruning before training, can achieve comparable performance to the state-of-the-art implementation of OMP. However, the gap between this SOTA pruning-at-initialization method and our method still exists. Besides, the result shows that EARLYBIRD can effectively achieve comparable or even better testing performance than OMP in most different

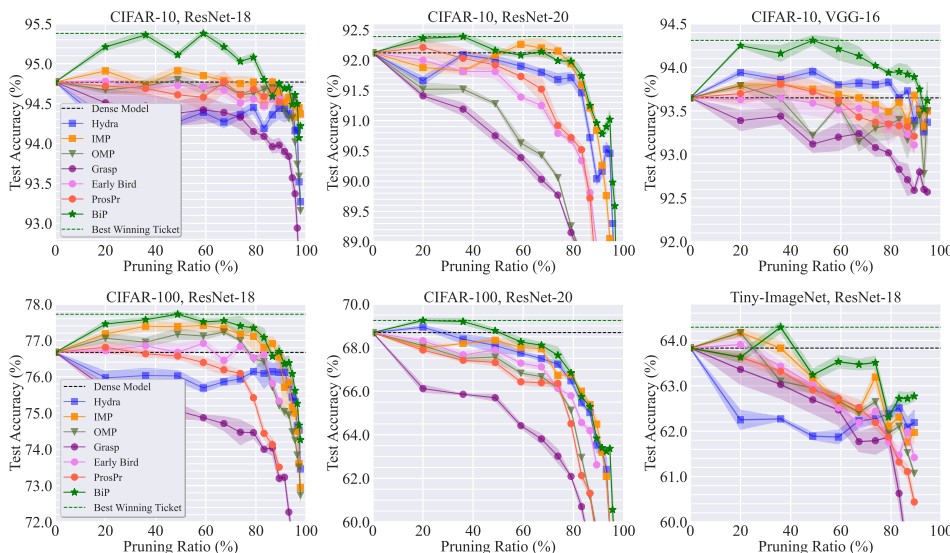

Figure A5: Unstructured pruning trajectory given by test accuracy (%) vs. pruning ratio (%). The visual presentation setting is consistent with Fig. 4. We consider two more baseline methods: EARLYBIRD [28] and PROSPR [25].

architecture-dataset combinations, which is also the main contribution of [28]. However, EARLYBIRD is still not as strong as IMP in testing performance.

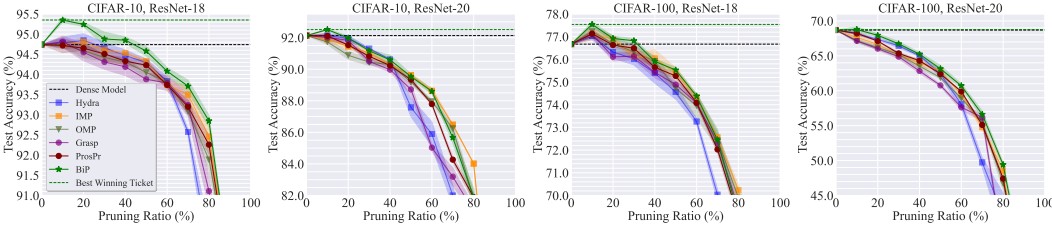

Figure A6: Structured pruning trajectory given by test accuracy (%) vs. pruning ratio (%). The visual presentation setting is consistent with Fig. 6. We add PROSPR [25] as our new baseline.

**Experiments on structured pruning with more baselines.** We compare our proposed method BIP to the new baseline PROSPR on different datasets and architectures in Fig. A6. on the structured pruning setting. As we can see, BIP consistently outperforms PROSPR and still stands top among all the baselines.

**More results on ImageNet.** In Fig. A7, we provide additional results on the dataset ImageNet with ResNet-18 in the unstructured pruning setting, in addition to the results of (ImageNet, ResNet-50) shown in Fig. 4. As we can see, the performance of BIP still outperforms the strongest baseline IMP and the same conclusion can be drawn as Fig. 4.

**Sanity check of BIP on specialized hyperparameters.** In Fig.A8, we show the sensitivity of BIP to its specialized hyperparameters at lower-level optimization, including the number of SGD iterations ($N$) in ($\boldsymbol{\theta}$-step), and the regularization parameter $\gamma$ in (1). Fig. A8(a) shows the test accuracy of BIP-pruned models versus the choice of $N$. As we can see, more SGD iterations for the lower-level optimization do not improve the performance of BIP. This is because in BIP, the $\boldsymbol{\theta}$-step is initialized by a pre-trained model which does not ask for aggressive weight updating. The best performance of BIP is achieved at $N \leq 3$. We choose $N = 1$ throughout the experiments as it is computationally lightest. Fig. A8(b) shows the performance of BIP by varying $\gamma$. As we can see, the choice of $\gamma \in \{0.5, 1\}$ yields the best pruning accuracy across all pruning ratios. If $\gamma$ is too small, the lack

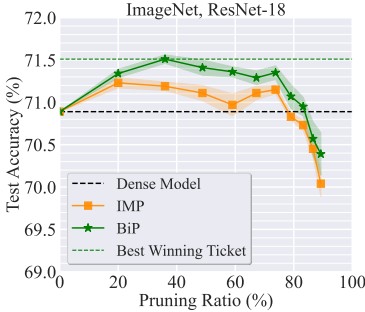

Figure A7: Unstructured pruning trajectory on ImageNet with ResNet-18. The experiment setting is consistent with Fig. 4. We only compare BIP with our strongest baseline IMP due to limited computational resource.

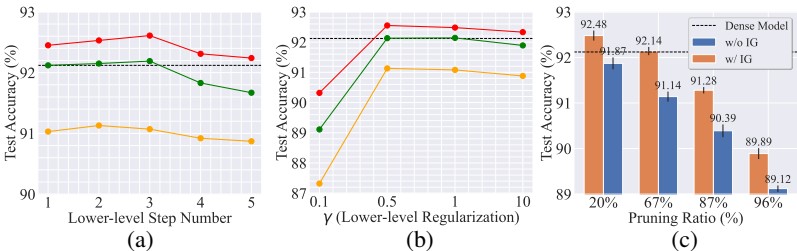

Figure A8: The sensitivity of BIP to (a) the lower-level step number $N$, (b) the lower-level regularizer $\gamma$, and (c) the contribution of the IG-term at various pruning ratios on (CIFAR-10, ResNet-20). Each curve or column represents a certain sparsity. In (c), we fix the $\gamma$ to 1.0 and compare the performance of the IG-involved/excluded (2) BIP.

of strong convexity of lower-level optimization would hamper the convergence. If $\gamma$ is too large, the lower-level optimization would depart from the true model objective and causes a performance drop. Fig. A8(c) demonstrates the necessity of the IG enhancement in BIP. We compare BIP with its IG-free version by dropping the IG term in (2). We observe that BIP without IG (marked in blue) leads to a significant performance drop ($> 1\%$) at various sparsities. This highlights that *the IG in the (**m**-step) plays a critical role in the performance improvements obtained by* BIP, justifying our novel BLO formulation for the pruning problem. In Fig. A9, we further demonstrate the influence of different choices of lower-level learning rate $\alpha$ as well as the batch size on the performance of BIP. Fig. A9 (a) shows that the test accuracy of BIP-pruned models is not quite sensitive to the choice of $\alpha \in \{0.01, 0.008\}$. A large $\alpha$ value (*e.g.*, $\alpha > 0.05$) will slightly decrease the performance of BIP. By contrast, a small $\alpha$ is preferred due to the fact that the model parameters are updated based on the pre-trained values. Fig. A9 (b) shows how the batch size influences the performance. As we can see,

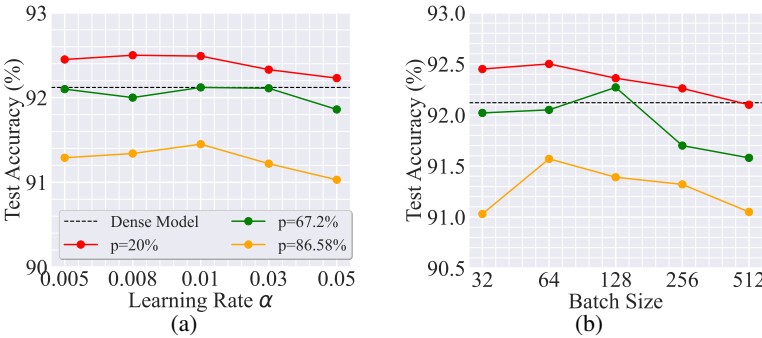

Figure A9: Ablation studies of BIP on different hyper-parameters. All the experiments are based on CIFAR-10 with ResNet-20. We select three sparsity values of a wide range (from not sparse to extreme sparse) to make the results more general. We study the influence of different (a) lower-level learning rate $\alpha$ and (b) batch size.

a large batch size might hurt the stochasticity of the algorithm and thus degrades the performance. We list our detailed batch size choices for different datasets in Tab. A1.

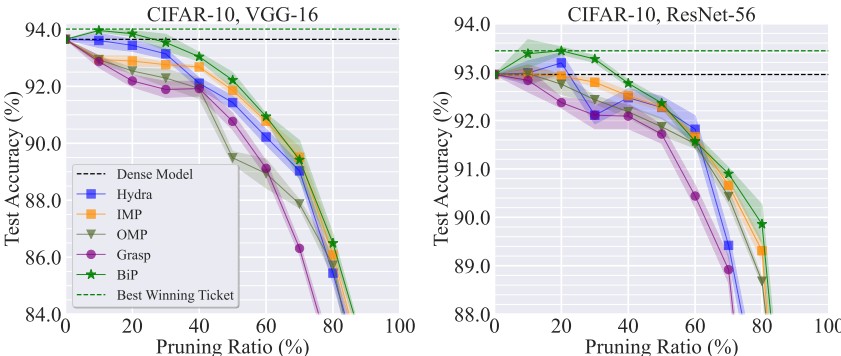

Figure A10: Filter-wise pruning test accuracy (%) v.s. sparsity (%) on CIFAR-10 with VGG-16 and ResNet-56.

**Additional structured pruning experiments.** In addition to filter pruning in Fig. 6, we provide more results in the structured pruning setting, including both filter-wise and channel-wise pruning (as illustrated in Fig. A3). In Fig. A10, results on CIFAR-10 with VGG-16 and ResNet-56 are added as new experiments compared to Fig. 6. Fig. A11 shows the results of the channel-wise pruning. As we can see, consistent with Fig. 6, BIP is able to find the best winning tickets throughout the experiments while it is difficult for IMP to find winning tickets in most cases. We also notice that HYDRA, as the optimization-based baseline, serves as a strong baseline in filter-wise pruning. It also indicates the superiority of the optimization-based methods over the heuristics-based ones in dealing with more challenging pruning settings.

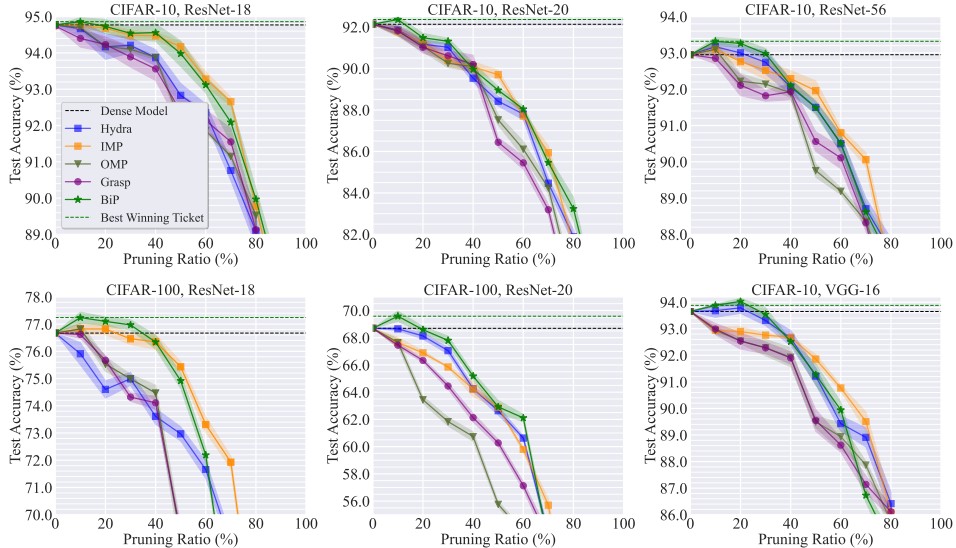

Figure A11: Channel-wise pruning test accuracy (%) v.s. sparsity (%). Settings are consistent with Fig. A10.

**Training trajectory of BIP.** We show in Fig. A12 that the BIP algorithm converges quite well within 100 training epochs using a cosine learning rate scheduler.

**The training trajectory of the mask IoU score.** To verify the argument that the mask also converges at the end of the training, we show the training trajectory of the mask similarity between two adjacent-epoch models in Fig. A13 at different pruning ratios. Here the mask similarity is represented through the intersection of the union (IoU) score of the two masks found by two adjacent epochs. The IoU score ranges from 0.0 to 1.0, and a higher IoU implies a larger similarity between

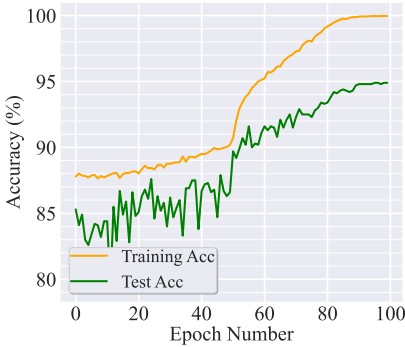

Figure A12: The training trajectory of BIP for unstructured pruning on (CIFAR-10, ResNet-18) with a pruning ratio of $p = 80\%$.

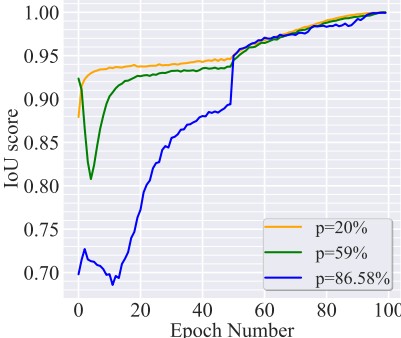

Figure A13: Training trajectory of the IoU (intersection over union) score between the masks of two adjacent epochs. We show the trajectory of different pruning ratios.

the two masks. As we can see, the IoU score converges to 1.0 in the end, which denotes that the mask also converges at the end of the training phase. Also, with a smaller pruning ratio, the mask turns to converge more quickly.

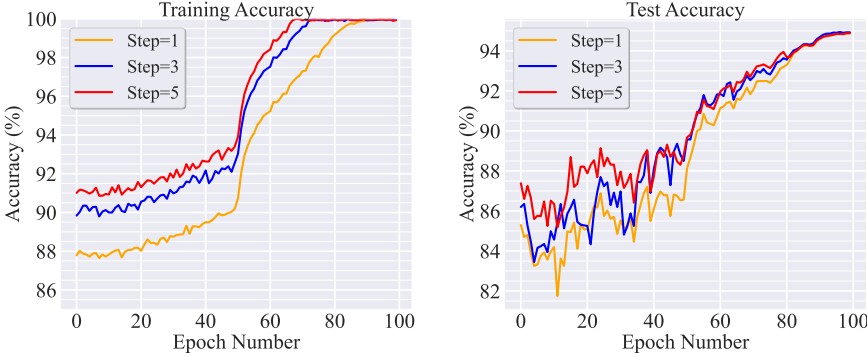

Figure A14: Training dynamics of BIP with different lower-level (SGD) steps on (CIFAR-10, ResNet-18) with the pruning ratio of p=80%.

**The effect of different lower-level steps in BIP on the training dynamics.** We conduct additional experiments to demonstrate the effectiveness of using one-step SGD in BIP. In our new experiments, we consider the number of SGD steps, 1, 3, and 5. We report the training trajectories of BIP in Fig. A14. As we can see, the use of multi-step SGD accelerates model pruning convergence at its early phase. Yet, if we run BIP for a sufficient number of epochs (we used 100 by default in other experiments), the final test accuracy of using different SGD settings shows little difference. Although

the use of multiple SGD steps could improve the convergence speed, it introduces extra computation complexity per BLO step. Thus, from the overall computation complexity perspective, using 1 SGD step but running more epochs is advantageous in practice.

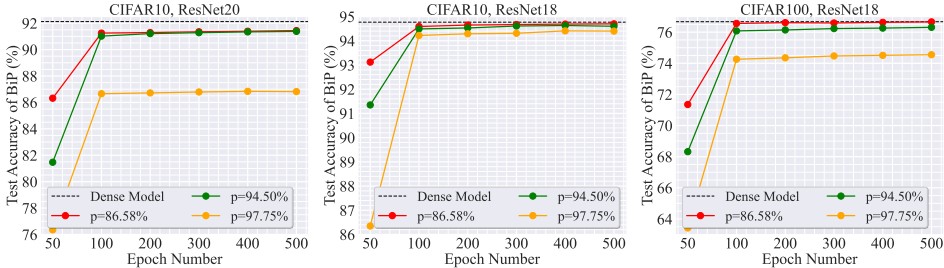

Figure A15: The effect of total training epoch number on the test accuracy with large pruning ratios. The epoch number is by default set to 100 in this paper. In each sub-figure, we report the performance of BIP with three different sparse ratios $p$.

**The effect of larger training epoch numbers with extreme sparsities.** We allow more time (training epochs) for BIP when a higher pruning rate is considered and the results are shown in Fig. A15. Specifically, we test three datasets and consider three pruning ratios (p=86.58%, 94.50%, 97.75%). For each pruning ratio, we examine the test accuracy of BIP versus the training epoch number from 50 to 500. Note that the number of training epochs in our original experiment setup was set to 100. As we can see, the performance of BIP gets saturated when the epoch number is over 100. Thus, even for a higher pruning ratio, the increase of training epoch number over 100 does not gain much improvement in accuracy.

**The effect of different training batch schemes.** We conducted ablation studies on three different schemes of BIP's training batches for the upper and lower level. In addition to two different random batches for the two levels, we also consider the same batch and the reverse batch scheme. BIP (same batch) always uses the same data batches for the two levels in each iteration while BIP (reverse batch) uses the data batches in a reversed order for the two level. Fig. A16 shows that both the random batch scheme (i.e., BIP) and the reverse batch scheme can bring a better testing accuracy performance than the same batch scheme throughout different pruning ratio settings. Fig. A17 shows that BIP the same batch scheme converges slower compared to the other two. Both of the results indicate BIP benefits from the diverse batch selection.

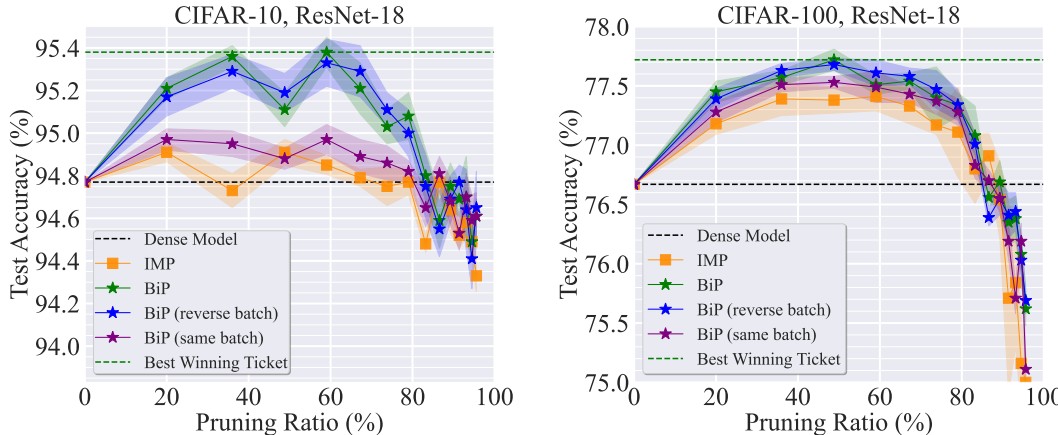

Figure A16: The effect of different training batch schemes on the performance of BIP. We consider two different variants of BIP denoted as BIP (reverse batch) and BIP (same batch). For BIP (reverse batch), the data batches are fed into the upper- and lower-step in a reversed order within each epoch, while for BIP (same batch), the data batches for upper- and lower-level are always the same. Experiment settings are consistent with Fig. 4. For better readability, we only plot the strongest baseline IMP for comparison.

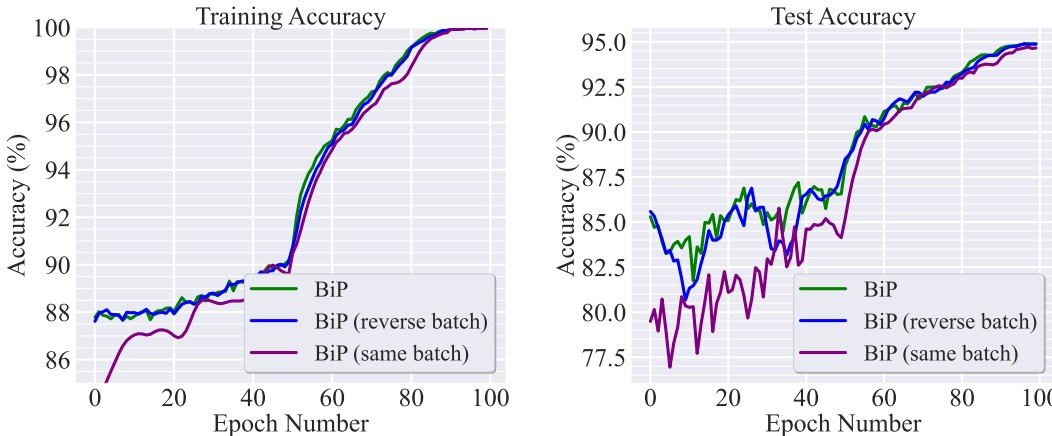

Figure A17: The effect of different training batch schemes on the training dynamics of BιP. We plot the training dynamics of different variants of BιP on (CIFAR-10, ResNet-18) with the pruning ratio of 80%.

Table A4: The sparsest winning tickets found by different methods on Tiny-ImageNet and ImageNet datasets. Winning tickets refer to the sparse models with an average test accuracy no less than the dense model [20]. In each cell, $p\%$ (acc±std%) represents the sparsity as well as the test accuracy. The test accuracy of dense models can be found in the header. ✗ signifies that no winning ticket is found by a pruning method. Given the data-model setup (*i.e.*, per column), the sparsest winning ticket is highlighted in **bold**.

| Method | Tiny-ImageNet ResNet-18 (63.83%) | ImageNet ResNet-18 (70.89%) | ResNet-50 (75.85%) |
|---|---|---|---|
| IMP | 20% (64.17±0.11%) | 74% (71.15±0.19%) | **80**% (76.05±0.13%) |
| OMP | 20% (64.17±0.11%) | ✗ | ✗ |
| GRASP | ✗ | ✗ | ✗ |
| HYDRA | ✗ | ✗ | ✗ |
| BιP | **36**% (64.29±0.13%) | **83**% (70.95±0.12%) | 74% (76.09±0.11%) |

## C.4 Broader Impact

We do not recognize any potential negative social impacts of our work. Instead, we believe our work can inspire many techniques for model compression. The finding of structure-aware winning ticket also benefits the design of embedded solutions to deploying large-scale models on resource-limited edge devices (*e.g.*, FPGAs), providing broader impact on both scientific research and practical applications (*e.g.*, autonomous vehicles).