# OpenReview forum: "Advancing Model Pruning via Bi-level Optimization"
_NeurIPS.cc/2022/Conference — NeurIPS 2022 Accept_

### Official Review · Reviewer_mQEK · 2022-07-06

**Rating:** 2
**Confidence:** 5
**Soundness:** 2 fair
**Presentation:** 1 poor
**Contribution:** 2 fair

**Summary:**

The authors (correctly) identify that iterative magnitude pruning (IMP) is inefficient to extract "winning tickets" from neural networks. To this end, they investigate a method utilising bi-level optimisation (BLO) to extract winning tickets -- including structured results which can easily yield real-world speedups. In the paper they formulate pruning as a BLO problem, and subsequently evaluate across CIFAR-10/100, Tiny-ImageNet and ImageNet. The results are consistently better than IMP, and gradient-based saliency methods such as Grasp.

==

See below for response to authors; score has been updated after rebuttal.

**Questions:**

Many of the questions are motivated from above:

1. What is the definition of "winning tickets" used?
2. Is the pruning done continuously through training?
3. What is the motivation for not using Frankle's benchmark?
4. How fast are the channel pruned networks?

**Limitations:**

I don't think there is any explicit discussion of limitations in the main paper. The checklist points us to appendix C4 but there's nothing there regarding limitations (and I have discussed some above).

**Strengths And Weaknesses:**

I am suspicious of the contributions of this paper, since the use of "winning ticket" seems to be overloaded in this work. In the abstract this term is defined as: "i.e., pruned sparse models with better generalization than the original dense models". But -- this is not the widely agreed upon definition. Lines 35 to 37 give the more widely accepted definition, but line 325 gives a different definition which I don't agree with. It is crucial that the work is self-consistent, and consistent with other literature.

Following on from this observation, it seems that the pruning is done throughout training. This cannot reasonably be compared to IMP or Grasp: the pruning mask is set at iteration 0 (or close to 0) and does not change throughout training. It is also worth noting that Grasp is not SOTA anymore: you should compare to ProsPr by Alizadeh et al. (cited by your work already). The only valid comparison I can see would be to the Early-Bird work by You et al. -- but you don't provide any in this direction. In addition, you have provided few comparisons to methods which prune after training, and perhaps do some small amount of fine tuning. These are also reasonable competitors to this work.

Finally, you are not the first to do bi-level optimisation for pruning. "Differentiable Network Pruning for Microcontrollers" by Liberis and Lane would be an example prior art (and I'm sure there are others).

Here are some more minor thoughts:

- Code is included which is very nice since the method is fairly complicated.
- I really commend this work for paying attention to structured pruning -- it's a really hard problem, and far more relevant than most other pruning directions -- but I am not sure you characterise the related work fairly. "Single Shot Structured Pruning" by van Amersfoort et al. and ProsPr by Alizadeh et al. both assessed this direction at initialisation.
- On L229 you say that we can assume $\nabla^2 l = 0$. I am not convinced this is actually true in practice: after all, this is how many earlier pruning saliency methods worked. However, I am not saying that this makes the work incorrect; approximations pop up everywhere in DL.
- Is there a good reason to use reference 18 as the benchmark rather than the benchmark provided by Frankle's missing the mark work? It is very extensive, and other works have built upon it in recent years.
- Figure 6a is not enough to justify the robustness to rewinding that marks a winning ticket. You've provided results on the smallest model with the easiest dataset. More difficult examples are needed.
- The final line of the conclusion is a bit concerning: in practice, structured pruning methods could prune the same number of parameters but yield totally different speedups. It is an important thing to measure.

---

> ### Author Response · Authors · 2022-08-02
> **Point-to-point Response to Reviewer mQEK (Part 3/3)**
>
> **Q10: Figure 6a is not enough to justify the robustness to rewinding that marks a winning ticket. You’ve provided results on the smallest model with the easiest dataset. More difficult examples are needed.**
>
> **A10:** Following the reviewer’s suggestion, we conduct additional experiments over more complex model-dataset combinations; see results in [Figure](https://ibb.co/Dzf5YQN). Our method is insensitive to rewinding epoch numbers, even for larger models (ResNet-18) or more complicated datasets (Tiny-ImageNet). A carefully tuned rewinding scheme does not lead to significant improvements, and thus, it is not necessary for BiP to rewind (and retrain) to achieve superior performance.
>
> **Q11: The final line of the conclusion is a bit concerning: in practice, structured pruning methods could prune the same number of parameters but yield totally different speedups. It is an important thing to measure. How fast are the channel pruned networks?**
>
> **A11:** Thanks for your suggestion. The main purpose of this paper is to advance the optimization foundation of the pruning problem through the lens of BLO. As the hardware acceleration is not the main purpose of this paper, we only mentioned this could be a future research direction to maximize the practical utility of BLO-enabled structured pruning to achieve hardware acceleration when compressing deep models at structural units, such as kernels/filters/channels.

---

> ### Author Response · Authors · 2022-08-02
> **Point-to-point Response to Reviewer mQEK (Part 2/3)**
>
> **Q3: It is also worth noting that Grasp is not SOTA anymore: comparisons with stronger initialization-based pruning methods, you should compare to ProsPr by Alizadeh et al. (cited by your work already)**
>
> **A3:** Following your suggestion, we conduct additional experiments to compare BiP with ProsPr in unstructured pruning (see [Figure](https://ibb.co/TbqXNFc)) and structured pruning (see [Figure](https://ibb.co/K0PTPnP)). The results show that ProsPr is indeed better than GraSP but is still not as good as IMP and our method in different architecture + dataset combinations. Meanwhile, except for the unstructured pruning settings of ResNet18 pruning over CIFAR10 and CIFAR100, ProsPr, as a pruning before training, can achieve comparable performance to the state-of-the-art implementation of  OMP [R11]. However, the gap between this SOTA pruning-at-initialization method and our method still exists. As a side note, we have covered more than one initialization-based baseline (SNIP, SynFlow) in Figure A3.
>
> > [R11] Ma, Xiaolong, et al. “Sanity checks for lottery tickets: Does your winning ticket really win the jackpot?.” Advances in Neural Information Processing Systems 34 (2021): 12749-12760.
>
> **Q4: The only valid comparison I can see would be to the Early-Bird work by You et al. – but you don’t provide any in this direction.**
>
> **A4:** Thank you for suggesting “the only valid comparison” with Early-Bird work [R11]. However, this might be a misunderstanding about our work. The suggested early-bird training in [R12] provides us with another one-shot pruning baseline. Additional experiments are conducted in [Figure](https://ibb.co/TbqXNFc) to compare with the early-bird training on 6 different settings. The [Figure](https://ibb.co/TbqXNFc) shows that early-bird training can effectively achieve comparable or even better testing performance than OMP in most different architecture+dataset combinations, which is also the main contribution of [R11]. However, the early-bird training is still not as strong as IMP in testing performance. Thus, we disagree that early-bird training is the only valid baseline for BiP. Please refer to [GR2](https://openreview.net/forum?id=t6O08FxvtBY&noteId=3pfCIFhqNF) for our clarification on the selection of baseline methods.
> > [R12] You, Haoran, et al. “Drawing early-bird tickets: Towards more efficient training of deep networks.” arXiv preprint arXiv:1909.11957 (2019).
>
> **Q5: In addition, you have provided few comparisons to methods which prune after training, and perhaps do some small amount of fine tuning.**
>
> **A5:** We respectfully disagree. Many of our baselines such as OMP and Hydra all prune after model training. This is in contrast to Grasp/SNIP/ProsPr, which are given by pruning at random initialization (i.e., before training).
>
> **Q6: You are not the first to do bi-level optimisation for pruning. “Differentiable Network Pruning for Microcontrollers” by Liberis and Lane would be an example of prior art (and I’m sure there are others).**
>
> **A6:** We respectfully disagree. Please refer to [GR1](https://openreview.net/forum?id=t6O08FxvtBY&noteId=3pfCIFhqNF).
>
> **Q7: “I am not sure you characterize the related work on structured pruning fairly. "Single Shot Structured Pruning" by van Amersfoort et al. and ProsPr by Alizadeh et al. both assessed this direction at initialisation.**
>
> **A7:** Thank you very much for pointing out these references. Following the reviewer’s suggestion, we conduct additional experiments to compare BiP with ProsPr in the context of structured pruning; see results in [Figure](https://ibb.co/K0PTPnP). As we can see, BiP consistently outperforms ProsPr and still stands top among all the baselines. During the rebuttal window, we were not able to add the comparison with **“Single Shot Structured Pruning”** as the codes are not released in the paper.  In the revision, both aforementioned papers will be cited and discussed in the related work.
>
> **Q8: On L229 you say that we can assume ∇2l=0. I am not convinced this is actually true in practice.**
>
> **A8:** As mentioned in Line 233-235 of our submission, this hessian-free assumption is not strict for ReLU-based neural networks (NNs) as the decision boundaries of NNs with ReLU activations are piecewise linear in a tropical hyper-surface [98]. And in practice, this is also a reasonable assumption and has been used in BLO-involved applications such as meta-learning [99] and adversarial learning [84].
>
> **Q9: Is there a good reason to use reference 18 as the benchmark rather than the benchmark provided by Frankle’s missing the mark work?**
>
> **A9:** We believe this is also a question related to baseline selection; see [GR2](https://openreview.net/forum?id=t6O08FxvtBY&noteId=3pfCIFhqNF). Frankle’s work is the benchmark of all different **one-shot pruning methods**. In contrast, reference [R11] provides the benchmark for the iterative magnitude pruning method.

---

> ### Author Response · Authors · 2022-08-02
> **Point-to-point Response to Reviewer mQEK (Part 1/3)**
>
> **Q1: The use of “winning ticket” seems to be overloaded in this work. What is the definition of “winning tickets” used? Lines 35 to 37 give the more widely accepted definition, but line 325 gives a different definition which I don’t agree with.**
>
> **A1:** This is a great comment. We apologize for the imprecise statement about winning tickets at Line 325. Yes, our definition of “winning tickets” follows Lines 35-37 but covers the early-epoch rewinding variant [R7] and the no-rewinding (i.e., fine-tuning) variant [R8] as special cases. To be more concrete,  a winning ticket $(\mathbf m, \boldsymbol \theta^\prime )$ is given by a pair of sparse mask $\mathbf m$ and model “initialization” $\boldsymbol \theta^\prime$, from which the non-sparse model weights are retrained to achieve the test accuracy greater than or equal to the test accuracy of the original dense model. It is worth noting that in the original LTH work [R9]  $\boldsymbol \theta^\prime$ was set by the random initialization $\boldsymbol \theta_0$ used in dense model training, namely, $\boldsymbol \theta^\prime =\boldsymbol \theta_0$ to realize the isolated training process. However, its follow-up work [R7] found that the early-epoch rewinding strategy (which sets $\boldsymbol \theta^\prime$  as an early-epoch dense model, i.e., $\boldsymbol \theta^\prime =\boldsymbol \theta_t$ for t-epoch training) typically yields the best test accuracy than the case of rewinding to the random initialization ($\boldsymbol \theta^\prime =\boldsymbol \theta_0$) and the case of no rewinding (i.e., $\boldsymbol \theta^\prime$ is set by the currently non-pruned model weights). Yet, rewinding has a downside as it takes additional computation costs besides pruning. We follow the above line of work to define our winning ticket $(\mathbf m, \boldsymbol \theta^\prime )$ to produce a subnetwork that can match or surpass the performance of the dense model. If the definition of winning tickets has to be aligned with the original LTH paper [R9], we could also use the notion of “matching subnetwork” to reflect the quality of a pruning method following Chen & Frankle’s work [R10].
>
> > [R7] Renda, Alex, Jonathan Frankle, and Michael Carbin. "Comparing rewinding and fine-tuning in neural network pruning." arXiv preprint arXiv:2003.02389 (2020).
> >
> > [R8] Chen, Tianlong, et al. "Long live the lottery: The existence of winning tickets in lifelong learning." International Conference on Learning Representations. 2020.
> >
> > [R9] Frankle, Jonathan, and Michael Carbin. "The lottery ticket hypothesis: Finding sparse, trainable neural networks." arXiv preprint arXiv:1803.03635 (2018).
> >
> > [R10] Chen, Tianlong, et al. “The lottery ticket hypothesis for pre-trained bert networks.” Advances in neural information processing systems 33 (2020): 15834-15846.
>
> **Q2: It seems that the pruning is done throughout training. This cannot reasonably be compared to IMP or Grasp: the pruning mask is set at iteration 0 (or close to 0) and does not change throughout training.**
>
> **A2:** Thank you for raising this great question. However, the comment “pruning is done throughout training” is not precise for our method (BiP). BIP is performed after dense model training as its initialization is given by the (pre-trained) dense model weights (see Line 282). This is also why we adopt the one-step gradient descent to realize model re-training (see Line 285 and Fig. 6(b1)).
>
> We agree that BIP involves the weight retraining process (i.e., the lower-level optimization task). But this is the same as IMP, which also requires weight re-training. In IMP, the $(t+1)$-th pruning round prunes the nonzero model weights that are retrained at the end of the $t$ pruning round. Thus, the pruning mask is also updated throughout training. In this sense, we feel that it is quite reasonable to compare IMP with BiP. And IMP gives an **upper bound** of the computation complexity of model pruning (see Line 109 - 123).
>
> Yes,  GraSP is the method of pruning at random initialization. Thus, the pruning mask is fixed and independent of training. However, we feel that it is also necessary to compare BiP with Grasp as the latter provides a **lower bound** of the computation complexity of model pruning.

---

> ### Author Response · Authors · 2022-08-02
> **General Response to Reviewer mQEK**
>
> Thank you very much for providing us with very constructive comments. In what follows, we begin by making a few general responses (GRs) based on the reviewer’s comments and then list our point-to-point answers to the raised questions.
>
> **GR1: Possible misunderstanding on our bi-level contribution.**
>
> Thanks for pointing out the missing reference [R1] “Differentiable Network Pruning for Microcontrollers” (by Liberis and Lane) and raising the question “you are not the first to do bi-level optimization for pruning (and I’m sure there are others)”. We will be sure to cite [R1] and discuss our **novelties** (vs. [R1]) in the revision. **Yet, we still believe** that a systematic study of BLO for model pruning was lacking in the literature, and ours is the first one in this direction.
>
> 1. Reference [R1] claimed using BLO for model pruning, but it refers to the **alternating optimization (AO)** procedure where pruning and training alternatively perform gradient descent.  Strictly speaking, this AO process does **NOT** exactly solve a BLO problem since it excludes the derivation of implicit gradient (IG) (see Line 217 - 230 in our submission). The IG challenge is a known problem in BLO; see the optimization literature [71] in our submission. To the best of our knowledge, we, for the first time, derived the closed form of IG for BLO-oriented pruning and showed that the bi-linearity of pruning variables makes the IG-involved gradient Eq. (2) easily solvable. The computational complexity is almost the same as that of computing the first-order gradient just once, as supported by Eq. (6). Our theoretical finding was summarized in Proposition 1. And Fig. 6-(b3) made a  sanity check for the importance of IG in BLO-oriented pruning. This is also a key difference from pruning methods that directly call for Darts-like formulation and approach (e.g. [R2], which was cited in [R1]), where the special BLO characteristic–bi-linearity of pruning variables–was not explored and exploited to simplify the IG computation.
> 2. The advantage of BLO for model pruning was not fully exploited in the existing literature (including [R1] and [R2]). We respectfully argue that we did not see any prior work to provide the explicit BLO interpretation of IMP and disentangle the non-sparse re-training from pruning using customizable lower-level optimization tasks; see our BLO formulation and its advantages in Line 189-216. **As Reviewer [iDDi](https://openreview.net/forum?id=t6O08FxvtBY&noteId=K-BISb3ND_a) pointed out**, “The proposed BIP pruning algorithm is original and practical.”
>
> > [R1] Liberis, Edgar, and Nicholas D. Lane. “Differentiable Network Pruning for Microcontrollers.” arXiv preprint arXiv:2110.08350 (2021).
> >
> > [R2] Ning, X., Zhao, T., Li, W., Lei, P., Wang, Y., and Yang, H. DSA: More efficient budgeted pruning via differentiable sparsity allocation. arXiv preprint arXiv:2004.02164, 2020.
>
> **GR2: Clarification of baseline method selection**
>
> The reviewer suggested several **one-shot/initialization-based baselines** (Early bird [R3], ProsPr [R4], Single-shot Structured Pruning [R5]) and questioned us why not consider the **one-shot pruning benchmark** ([R6] “Pruning Neural Networks at Initialization: Why Are We Missing the Mark?” by Frankle). Based on those comments, we feel that the reviewer might **mistakenly regard our proposed method as another one-shot or initialization-based pruning method**. This is not the main purpose of our work. Our goal is to seek the proper optimization basis for successful model pruning that can attain high pruned model accuracy (like IMP) without incurring a high computation cost as the model sparsity increases (namely, enjoying computation efficiency like one-shot pruning). Thus, performance-wise,  **IMP is our strongest and the main baseline throughout the experiments**. Meanwhile, we also consider comparing BiP with one-shot pruning since the latter gives a lower bound on the computation complexity of model pruning. **We also conducted additional experiments based on the reviewer's suggestion to enrich our baseline methods.** However, the conclusion is consistent: BiP outperforms all the newly added baselines (see [the summary of experiments](https://openreview.net/forum?id=t6O08FxvtBY&noteId=6LCTzm0sycR)).
>
> > [R3] You, Haoran, et al. “Drawing early-bird tickets: Towards more efficient training of deep networks.” arXiv preprint arXiv:1909.11957 (2019).
> >
> > [R4] Alizadeh, Milad, et al. “Prospect pruning: Finding trainable weights at initialization using meta-gradients.” arXiv preprint arXiv:2202.08132 (2022).
> >
> > [R5] van Amersfoort, Joost, et al. “Single shot structured pruning before training.” arXiv preprint arXiv:2007.00389 (2020).
> >
> > [R6] Frankle, Jonathan, et al. “Pruning neural networks at initialization: Why are we missing the mark?.” arXiv preprint arXiv:2009.08576 (2020).

---

> ### Author Response · Authors · 2022-08-02
> **Summary of Additional Experiments**
>
> We have conducted a series of new experiments based on the reviewer's comments. For ease of reading, we summarize them below.
> 1. For unstructured pruning settings, we add Early-Bird(one-shot pruning baseline) and ProsPr (initialization-based baseline) to the CIFAR-10, CIFAR-100, and TinyImageNet datasets (6 model architecture + dataset combinations). The results can be found in [Figure](https://ibb.co/TbqXNFc). A detailed discussion can be found in Q3&A3.
> 2. For structured pruning settings, we add ProsPr as the latest initialization-based baseline (4 model architecture + dataset combinations). The results can be found in [Figure](https://ibb.co/K0PTPnP). A detailed discussion can be found in Q7&A7.
> 3. To verify the insensitivity of BiP to rewinding epochs in more complicated settings, we conducted additional experiments on 3 more dataset-model architecture combinations (ResNet18 + CIFAR-10, ResNet18 + CIFAR-100, ResNet18 + TinyImageNet). The results can be found in [Figure](https://ibb.co/Dzf5YQN). More detailed discussions can be found in Q10&A10.
>
> We will include the additional experiments' results in the revised version.

---

> ### Author Response · Authors · 2022-08-05
> **Look forward to post-rebuttal feedback!**
>
> Dear Reviewer mQEK,
>
> Thank you very much for sparing your time to review our paper. In the posted response, we have tried our best to (1) clarify possible misunderstandings regarding the novelty of our work (see [General Response Link](https://openreview.net/forum?id=t6O08FxvtBY&noteId=3pfCIFhqNF) ), (2) conduct a series of additional experiments requested in the comments (see [summary of added experiments](https://openreview.net/forum?id=t6O08FxvtBY&noteId=6LCTzm0sycR)), and (3) address your questions point by point (see [Part I](https://openreview.net/forum?id=t6O08FxvtBY&noteId=xK2RVWyjAna), [Part II](https://openreview.net/forum?id=t6O08FxvtBY&noteId=axWLRoemTAz), [Part III](https://openreview.net/forum?id=t6O08FxvtBY&noteId=_HzCMw77WGT) respectively).
>
> We hope that you can find our effortful response convincing. If you have additional comments, please feel free to let us know. We will try our best to address them.

---

> ### Author Response · Authors · 2022-08-08
> **Two days left for open discussion**
>
> Dear Reviewer mQEK:
>
> We are very grateful to your constructive comments. We have made a substantial effort in responding to your questions and listed all the paper revisions and additional experiments in the **[summary of paper revisions and additional experiments](https://openreview.net/forum?id=t6O08FxvtBY&noteId=ckz79KyIMNi)**. As there are only two days left for author-reviewer discussions, we sincerely hope that you could provide us feedback before the discussion phase ends, and are happy to answer any follow-up questions.
>
> Best regards,
>
> Authors

---

> > ### Comment · Reviewer_mQEK · 2022-08-08
> > **Response to authors**
> >
> > GR1:
> >
> > I will accept the novelty regarding IGs, but it is important to be specific and contextualise your contributions
> >
> > GR2:
> >
> > See below. The framing of this work strikes me as extremely confusing
> >
> > Q1/A1:
> >
> > Your definition of winning tickets is far away from the standard definition used by the literature. Taking a pre-trained model and pruning it is not usually within the realm of the lottery ticket hypothesis. It is even less clear to me why you have chosen to frame your work this way. There's nothing wrong with simply having a technique that prunes pre-trained models -- but the casual reader who isn't as familiar with this part of the literature will be confused that this is actually what you are doing.
> >
> > Q2/A2:
> >
> > See answer to previous question.
> >
> > Your justification for why you are comparing your method -- which prunes a pre-trained model -- with GraSP / ProsPr which prune a random initialisation, is unconvincing. It is not sufficiently well explained in the paper. I don't see how they can be meaningfully compared with the additional context provided by the rebuttal.
> >
> > Q4/A4:
> >
> > It is not really surprising that early bird is not competitive with your work: it is pruning throughout training, but as you have clarified in the rebuttal, you are actually pruning pre-trained models.
> >
> > Q8/A8:
> >
> > I agree that this was a sensible assumption in my original review. However, my point was to suggest that you need to justify it better in the text as it's not strictly true.
> >
> > Q11/A11:
> >
> > This seems like a bit of an excuse. In theory timing the models is not a large amount of work -- I've done it myself for other works. It remains an important thing to assess: if only a small number of parameters are pruned early in the model then there will be only minor speedup. You could provide this information in a future version of this work.
> >
> > ====
> >
> > I am actually going to lower my score and argue for rejection on the basis that the paper's framing is deeply confusing. I am experienced and published in this field, and I couldn't follow what the paper was doing. The submission cannot solely be rated on its technical merits but also on how it is written and how the community will benefit from its publication.
> >
> > The conflation of what a "lottery ticket" is deeply concerning to me. Taking pre-trained dense models and assessing the lottery ticket hypothesis in this context is far from the norm in the literature. I am not convinced it is interesting to the community either.
> >
> > My issue is primarily with the writing and framing. I do believe that the authors may be able to write a more convincing and easy to follow paper if they actually focused solely on post-training pruning, and clearly compared to the state of the art baselines in this area. Magnitude pruning, for example, is an old baseline.
> >
> > This is a case of interesting work combined with confusing writing. At other venues I review for I would ask for accept with major revisions -- which I would want to see -- but this is not an option at NeurIPS. As such, I will strongly argue for rejection.

---

> > > ### Author Response · Authors · 2022-08-09
> > > **Respectfully but strongly disagree (Part 3/3)**
> > >
> > > **Q5: I agree that this was a sensible assumption in my original review. However, my point was to suggest that you need to justify it better in the text as it's not strictly true.**
> > >
> > > A5: We thank the reviewer for the clarification. We believe we had already discussed the justification for the hessian-free assumption in our original submission (line 232-235 in the current revision). Our response to you reiterated the same.
> > >
> > > **Q6: I am experienced and published in this field, and I couldn't follow what the paper was doing. I do believe that the authors may be able to write a more convincing and easy to follow paper.**
> > >
> > > A6: We have never doubted that the reviewer is well experienced and published in this field, and in fact we believe that all our reviewers are well qualified to review our paper.  We would like to point out that the other three reviewers rate the presentation of our work with a 3 (Reviewer [cu5X](https://openreview.net/forum?id=t6O08FxvtBY&noteId=cQiBjp_bI0P)), 4 (Reviewer [c9rK](https://openreview.net/forum?id=t6O08FxvtBY&noteId=OX68iAHfmv)), 4(Reviewer [iDDi](https://openreview.net/forum?id=t6O08FxvtBY&noteId=K-BISb3ND_a)). Reviewer cr9K summarizes our paper as “The Paper is concise and well written.” and “The theory is easy to follow.” Reviewer iDDi appreciates our writing with the comment, “The paper is well organized and easy to follow.”  We really do not think that our paper should be scored by “1, poor presentation”.
> > >
> > > **Q7:  I am not convinced it is interesting to the community either.**
> > >
> > > A7: It is unfortunate that the reviewer feels like this work is not interesting to the community. Once again, we would appreciate it if the reviewer could be specific about your reason for not being convincing.
> > >
> > > Since we do not know what makes the reviewer unconvincing, we cannot really address your concern. We can only cite other reviewers’ comments and high ratings to support our claim of novelty and contributions. As a side note, we have made a substantial effort and tried our best to address each of your previous questions (e.g., the IG-related one you accepted). If there is a positive point, we hope the reviewer could give this credit to our work.
> > >
> > > **Q8: I am actually going to lower my score and argue for rejection on the basis that the paper's framing is deeply confusing.**
> > >
> > > A8: We are disappointed that the reviewer feels so negatively about our work. We have tried our best to address the concerns the reviewer had in the initial review and addressed the novelty concern (as the reviewer themselves acknowledged), and added additional experiments suggested by the reviewer (with our proposed scheme being significantly better).
> > >
> > > It is confusing to us that addressing the reviewer's comments led them to score our paper as a "Strong Reject", which as per reviewer guidance, is defined as "a paper with major technical flaws, and/or poor evaluation, limited impact, poor reproducibility and mostly unaddressed ethical considerations". It is not clear from our discussion that we have "major technical flaw", "poor evaluation", "limited impact", "reproducibility", or "unaddressed ethical considerations" in our paper. If there is, again we would appreciate the reviewer to be specific about it.

---

> > > > ### Comment · Reviewer_mQEK · 2022-08-10
> > > > **Response to authors**
> > > >
> > > > > Since novelty of contributions was listed as one of the initial weaknesses
> > > >
> > > > No, that wasn't in my review or my response. I will not raise the contribution score unless I am satisfied that the work has been evaluated correctly.
> > > >
> > > > > These variants are well documented and exactly follow the line of research on LTH.
> > > >
> > > > I agree.
> > > >
> > > > > We would like to stress that the definition of winning tickets is evolving with time and is never rigid or constrained to a specific paper, especially in the development of different initialization strategies of non-zero model weights given a pruning mask [R1-R4]. Therefore, we disagree that our definition of winning tickets is “far away” from the literature.
> > > >
> > > > It is misleading to suggest that finding lottery tickets for a pre-trained model is as much as an achievement as it is for randomly initialised models. It has limited practical value, because the primary value of the LTH if we can exploit it in practice is that we can try to reduce the cost of training models. Taking pretrained models and pruning them does not yield this benefit.
> > > >
> > > > This is crucial. If you have a method for pruning after training -- which is what you do have -- then you should evaluate it like one.
> > > >
> > > > > The original LTH paper also pruned the model starting from pre-trained weights.
> > > >
> > > > Following on from the previous point, the original iterative magnitude pruning method is not supposed to be a "practical" method in any real sense. It is proposed as a method which gets acceptable results, demonstrating the existence of lottery tickets, but without necessarily being practical. It is not a practical benchmark in any sense other than accuracy.
> > > >
> > > > >  We also respectfully bring the reviewer’s attention to the fact that all the other reviewers rated our presentation either “good” or “excellent”. We strongly disagree that writing is a weakness of this work, although we are unfortunate to learn that the reviewer rated our submission with only a score of 1 (poor presentation).
> > > >
> > > > I am not required to agree with the other reviewers on everything. If we always did, then there would be no point in having multiple reviewers. Other reviewers will also have a different research background to me, which will naturally lead us to have different perspectives on this work.
> > > >
> > > > > comparing methods that prune at random initialization, such as GraSP, was very common in evaluating pruning methods that start from pre-trained models; please see [R3, R7], and Table 2 and Table 4 in [R5] for comparing with the IMP.
> > > >
> > > > There are two points to be clear about here: 1) just because someone else has done it doesn't make it right. And 2) I am not even suggesting that you don't include the comparisons, but that you clearly categorise the baselines and make it clear why they may / may not beat your method. It's not clear as currently written.
> > > >
> > > > > Finally, please note that the ProsPr [R6] baseline was suggested by the reviewer in your original comment, and we thus added this to our experiment. Therefore, we hope that we will not be criticized by comparing ourselves with ProsPr.
> > > >
> > > > I don't plan to criticise you for including proper baselines. Per my comment above, you should accurately categorise the competing methods.
> > > >
> > > > > A5: We thank the reviewer for the clarification. We believe we had already discussed the justification for the hessian-free assumption in our original submission (line 232-235 in the current revision). Our response to you reiterated the same.
> > > >
> > > > When you read my review again you will see that I was not even complaining about this assumption. I agree it is reasonable. I hoped you would find the additional context useful if you weren't already aware of it.
> > > >
> > > > > We would like to point out that the other three reviewers rate the presentation of our work
> > > >
> > > > See comment above. I am not required to agree at this stage. I respectfully disagree with their opinion, and I am allowed to. My complains regarding the evaluation of your work and meaningful comparison to baselines remain.
> > > >
> > > > > Once again, we would appreciate it if the reviewer could be specific about your reason for not being convincing.
> > > >
> > > > See above regarding the utility of pruning methods that prune before, during, and after training.
> > > >
> > > > > It is confusing to us that addressing the reviewer's comments led them to score our paper as a "Strong Reject"
> > > >
> > > > Poor evaluation and limited impact.
> > > >
> > > > ====
> > > >
> > > > I'll stay at my score.

---

> > > > > ### Author Response · Authors · 2022-08-10
> > > > > **Further Response to Reviewer mQEK**
> > > > >
> > > > > **Q1: Since novelty of contributions was listed as one of the initial weaknesses — No, that wasn't in my review or my response.**
> > > > >
> > > > > A1: The comment
> > > > >
> > > > > > Finally, you are not the first to do bi-level optimisation for pruning. "Differentiable Network Pruning for Microcontrollers" by Liberis and Lane would be an example prior art (and I'm sure there are others).
> > > > >
> > > > >  is what we understood as a criticism to our technical contribution. We responded to this comment, and you accepted the novelty regarding IG.
> > > > >
> > > > > **Q2: Finding Lottery tickets for a pre-trained model has limited practical value. The primary value of LTH is to try to reduce the cost of training models. The original IMP is not supposed to be practical in any real sense. It is not a practical benchmark in any sense other than accuracy.**
> > > > >
> > > > > A2: We have a different understanding on “winning tickets”. Yes, the trainability is a great merit of the original LTH, while the quality of pruning (finding subnetwork with improved generalization) is a more important property to us and should not be omitted. For example, LTH offers sparsity which not only maintains the great generalization ability but also benefits striking a graceful balance with other performance metrics, e.g., OOD robustness [R1] and transfer learning ability [R2]. IMP, albeit being very computationally expensive, is still the scheme that finds subnetworks with the highest accuracies. Our proposal offers a new optimization basis to find subnetworks as accurate or more than IMP but significantly more efficient than IMP. Those points have been clearly stated in the paper and our previous responses.
> > > > >
> > > > > > [R1] Diffenderfer, James, et al. “A winning hand: Compressing deep networks can improve out-of-distribution robustness.” Advances in Neural Information Processing Systems 34 (2021): 664-676.
> > > > > >
> > > > > > [R2] Chen, Tianlong, et al. “The lottery tickets hypothesis for supervised and self-supervised pre-training in computer vision models.” Proceedings of the IEEE/CVF Conference on Computer Vision and Pattern Recognition. 2021.
> > > > >
> > > > > **Q3: Clearly categorize the baselines and make it clear why they may / may not beat your method. It's not clear as currently written.**
> > > > >
> > > > > A3: From our point of view, we did categorize our pruning baselines and provided the rationale behind them. As shown in Sec. 2, those one-shot, initialization-based pruning methods are motivated and incorporated from the computation efficiency perspective; see our response on [the full spectrum of the computational efficiency and quality trade-off](https://openreview.net/forum?id=t6O08FxvtBY&noteId=KV5uNe2P1f).
> > > > >
> > > > > **Q4: I am not required to agree with the other reviewers on everything.**
> > > > >
> > > > > A4: Citing the other reviewers’ assessment was not to force the reviewer to have to agree with them. Instead, we aim to provide justifications like other reviewers why the score of **1 poor** for presentation does not sound like a quite fair rating for our work.

---

> > > > ### Author Response · Authors · 2022-08-10
> > > > **Further Response to Reviewer mQEK**
> > > >
> > > > Thank you for posting [Response to authors](https://openreview.net/forum?id=t6O08FxvtBY&noteId=Q1S0FPLOen_). Please see our follow-up clarification and response below.
> > > >
> > > > **Q1: Since novelty of contributions was listed as one of the initial weaknesses — No, that wasn't in my review or my response.**
> > > >
> > > > A1: The comment
> > > >
> > > > > Finally, you are not the first to do bi-level optimisation for pruning. "Differentiable Network Pruning for Microcontrollers" by Liberis and Lane would be an example prior art (and I'm sure there are others).
> > > >
> > > >  is what we understood as a criticism to our technical contribution. We responded to this comment, and you accepted the novelty regarding IG.
> > > >
> > > > **Q2: Finding Lottery tickets for a pre-trained model has limited practical value. The primary value of LTH is to try to reduce the cost of training models. The original IMP is not supposed to be practical in any real sense. It is not a practical benchmark in any sense other than accuracy.**
> > > >
> > > > A2: We have a different understanding on “winning tickets”. Yes, the trainability is a great merit of the original LTH, while the quality of pruning (finding subnetwork with improved generalization) is a more important property to us and should not be omitted. For example, LTH offers sparsity which not only maintains the great generalization ability but also benefits striking a graceful balance with other performance metrics, e.g., OOD robustness [R1] and transfer learning ability [R2]. IMP, albeit being very computationally expensive, is still the scheme that finds subnetworks with the highest accuracies. Our proposal offers a new optimization basis to find subnetworks as accurate or more than IMP but significantly more efficient than IMP. Those points have been clearly stated in the paper and our previous responses.
> > > >
> > > > > [R1] Diffenderfer, James, et al. “A winning hand: Compressing deep networks can improve out-of-distribution robustness.” Advances in Neural Information Processing Systems 34 (2021): 664-676.
> > > > >
> > > > > [R2] Chen, Tianlong, et al. “The lottery tickets hypothesis for supervised and self-supervised pre-training in computer vision models.” Proceedings of the IEEE/CVF Conference on Computer Vision and Pattern Recognition. 2021.
> > > >
> > > > **Q3: Clearly categorize the baselines and make it clear why they may / may not beat your method. It's not clear as currently written.**
> > > >
> > > > A3: From our point of view, we did categorize our pruning baselines and provided the rationale behind them. As shown in Sec. 2, those one-shot, initialization-based pruning methods are motivated and incorporated from the computation efficiency perspective; see our response on [the full spectrum of the computational efficiency and quality trade-off](https://openreview.net/forum?id=t6O08FxvtBY&noteId=KV5uNe2P1f).
> > > >
> > > > **Q4: I am not required to agree with the other reviewers on everything.**
> > > >
> > > > A4: Citing the other reviewers’ assessment was not to force the reviewer to have to agree with them. Instead, we aim to provide justifications from other reviews why the score of **1 poor** for presentation does not seem a quite fair rating to our work.

---

> > > ### Author Response · Authors · 2022-08-09
> > > **Respectfully but strongly disagree (Part 2/3)**
> > >
> > > **Q3: It is even less clear to me why you have chosen to frame your work this way.**
> > >
> > > A3: It is not clear which aspect of our work the reviewer is not clear about. We would appreciate it if the reviewer could be specific about what “frame your work this way” is referring to.
> > >
> > > We have provided a detailed description of the problem and our motivation in Section 2. We wish to advance the optimization basis of model pruning for obtaining high pruning accuracy while being significantly more computationally efficient. To develop such a pruning scheme, we leverage the bilevel framework and show that pruning is a very special class of bilevel optimization problem that can be solved accurately and efficiently. We also respectfully bring the reviewer’s attention to the fact that all the other reviewers rated our presentation either “good” or “excellent”. We strongly disagree that writing is a weakness of this work, although we are unfortunate to learn that the reviewer rated our submission with only a score of 1 (poor presentation).
> > >
> > >
> > > **Q4: Your justification for why you are comparing your method -- which prunes a pre-trained model -- with GraSP / ProsPr which prunes a random initialisation, is unconvincing. It is not sufficiently well explained in the paper. It is not really surprising that early bird is not competitive with your work: it is pruning throughout training, but as you have clarified in the rebuttal, you are actually pruning pre-trained models.**
> > >
> > > A4: We respectfully disagree with this comment.
> > >
> > > First, we would like to make it very clear that, comparing methods that prune at random initialization, such as GraSP, was very common in evaluating pruning methods that start from pre-trained models; please see [R3, R7], and Table 2 and Table 4 in [R5] for comparing with the IMP. This is because GraSP is a representative pruning method with the least computation complexity, and it also aims to find winning tickets;  see our motivation section Sec 2 and Fig. 2. Therefore, this practice does not hurt the validity of our approach, but only supports it, as has been argued in aforementioned works.
> > >
> > > Second, we have  **not only** compared with methods such as GraSP, but have also compared with methods such as IMP and Hydra, which prune from **pretrained** models. By putting all these methods together, we hope to provide a full spectrum of the computational efficiency and quality trade-off of different types of model pruning methods; see the comparison with BiP in Fig. 5.
> > >
> > > Third, about the early-bird approach, we have made it very clear in our initial submission (before the rebuttal; see Line 282), since the proposed BiP approach starts from a pre-trained model like IMP, thus, the early-bird approach was not included in our initial submission because we know that comparing with it would be unfair. Knowing that such a comparison is unfair, we nevertheless add it in the revised manuscript **at the request of the reviewer**, and the new results just confirmed our earlier intuition. However, the reviewer seems to criticize us because we followed your request?
> > >
> > > Finally, please note that the ProsPr [R6] baseline was suggested by the reviewer in your original comment, and we thus added this to our experiment. Therefore, we hope that we will not be criticized by comparing ourselves with ProsPr.
> > >
> > >
> > > > [R5] Wang, Chaoqi, Guodong Zhang, and Roger Grosse. "Picking winning tickets before training by preserving gradient flow." arXiv preprint arXiv:2002.07376 (2020).
> > > >
> > > > [R6] Alizadeh, Milad, et al. "Prospect pruning: Finding trainable weights at initialization using meta-gradients." arXiv preprint arXiv:2202.08132 (2022).
> > > >
> > > > [R7] Chen, Xiaohan, et al. "The elastic lottery ticket hypothesis." Advances in Neural Information Processing Systems 34 (2021): 26609-26621.

---

> > > ### Author Response · Authors · 2022-08-09
> > > **Respectfully but strongly disagree (Part 1/3)**
> > >
> > > We thank the reviewer very much for your response, but we strongly disagree with the current criticism and the reasons for lowering the score.
> > >
> > > **Q1: I will accept the novelty regarding IGs, but it is important to be specific and contextualize your contributions**
> > >
> > > A1: It is encouraging for authors to see that the reviewer accepts the novelty of IG. In fact, our response and submission (Lines 211-216 and the section “Optimization foundation of BIP”) have clearly described the technical challenge of IG and our technical novelty. Since **novelty of contributions** was listed as one of the initial weaknesses, we would appreciate it if the contribution score was updated to reflect the fact that we could address the concern regarding the novelty of contributions.
> > >
> > >
> > > **Q2: Your definition of winning tickets is far away from the standard definition used by the literature. The conflation of what a "lottery ticket" is deeply concerning to me. Taking pre-trained dense models and assessing the lottery ticket hypothesis in this context is far from the norm in the literature.**
> > >
> > > A2: We respectfully disagree with this comment.
> > >
> > > **First**, we do not think we have conflated the idea of "lottery ticket" in our paper. To our best understanding, the reviewer’s concern remains in the term “winning ticket” (note that we did not use the term “lottery ticket” in our paper). As we have replied in our [previous response](https://openreview.net/forum?id=t6O08FxvtBY&noteId=xK2RVWyjAna), the definition of the “winning ticket” covers the early-epoch rewinding variant [R1] and the no-rewinding (i.e., finetuning) variant [R2] as special cases. These variants are well documented and exactly follow the line of research on LTH. We would like to stress that the definition of winning tickets is evolving with time and is never rigid or constrained to a specific paper, especially in the development of different initialization strategies of non-zero model weights given a pruning mask  [R1-R4]. Therefore, we disagree that our definition of winning tickets is “far away” from the literature.
> > >
> > > **Second**, we disagree with the comment, “Taking pre-trained dense models and assessing the lottery ticket hypothesis in this context is far from the norm in the literature.” The original LTH paper also pruned the model starting from pre-trained weights. This always occurs at the first pruning stage of IMP used in LTH. To justify our argument, please refer to [the repo of IMP for LTH contributed by Jonathan Franckle](https://github.com/google-research/lottery-ticket-hypothesis) or [IMP for LTH implemented in PyTorch](https://github.com/rahulvigneswaran/Lottery-Ticket-Hypothesis-in-Pytorch/). In the first repo,  please see  [Line 61](https://github.com/google-research/lottery-ticket-hypothesis/blob/1f17279d282e729ee29e80a2f750cfbffc4b8500/foundations/experiment.py#L61). In the second repo, please see [Line119-120](https://github.com/rahulvigneswaran/Lottery-Ticket-Hypothesis-in-Pytorch/blob/34a8c9678406a1c7dd0fec4c9f0d25d017be55fb/main.py#L119). In both repos, the pruning starts from the pre-trained model.
> > >
> > > It is unconvincing to us that the reviewer claimed that our strategy is “far from” the norm in the literature but without providing any evidence of such a claim.
> > >
> > > >[R1] Renda, Alex, Jonathan Frankle, and Michael Carbin. "Comparing rewinding and fine-tuning in neural network pruning." arXiv preprint arXiv:2003.02389 (2020).
> > > >
> > > >[R2] Chen, Tianlong, et al. "Long live the lottery: The existence of winning tickets in lifelong learning." International Conference on Learning Representations. 2020.
> > > >
> > > >[R3] Chen, Tianlong, et al. "Coarsening the granularity: Towards structurally sparse lottery tickets." arXiv preprint arXiv:2202.04736 (2022).
> > > >
> > > >[R4] Chen, Tianlong, et al. “The lottery ticket hypothesis for pre-trained bert networks.” Advances in neural information processing systems 33 (2020): 15834-15846.

---

### Official Review · Reviewer_iDDi · 2022-07-11

**Rating:** 8
**Confidence:** 4
**Soundness:** 3 good
**Presentation:** 4 excellent
**Contribution:** 4 excellent

**Summary:**

This paper provides a novel reformulation of model pruning as a bi-level optimization (BLO) problem, in which the paradigm of pruning-retraining pruning can be viewed as two optimization levels: (1) finding the pruning mask (the upper-level), and (2) masked model retraining (the lower-level).

The paper further proposed an algorithm, bi-level pruning (BIP), to be a BLO solver that uses only the first-order gradient, which makes it as efficient as one-shot pruning.

The experiment results show that BIP equips the high efficiency of one-shot pruning and maintains the high accuracy of iterative magnitude pruning (IMP) in both structured and unstructured pruning schemes.

**Questions:**

- It would be really helpful to add some discussion about the similarities and differences between the BLO and the L0-based pruning (https://arxiv.org/pdf/1712.01312.pdf).
- In Figure 6(b1) the best accuracy is achieved when lower-level steps N <= 3, which would be unexpected and need some explanation. As a larger N tends to find more optimal model parameters that are close to $\theta^*$.
- Could you talk about the schedule of BIP to prune with multiple sparsity ratios? IMP could initialize from models with low pruning ratios to produce models with high pruning ratios. Thus when pruning for multiple ratios, the time complexity of IMP could be amortized. I wonder if BIP could be efficient in this setting.

**Limitations:**

The authors have addressed the limitations adequately. It would be better to move some ablation studies like Figure A4 to the main manuscript, though the pages may be limited.

**Strengths And Weaknesses:**

Strengths:

- The idea of the BLO reformulation of the model pruning problem is new, which provides a theoretical basis for BLO algorithms to be explored for model pruning.
- The proposed BIP pruning algorithm is original and practical.
- The paper is well organized and easy to follow.

Weakness:

- It would be really helpful to add some discussion about the similarities and differences between the BLO and the L0-based pruning (https://arxiv.org/pdf/1712.01312.pdf). L0-based methods view pruning masks as random variables defined by specific parameters, thus making the pruning masks could be learned together with the models. This is a different perspective compared to BLO. And these two views may be combined and unified, which makes it worth having a discussion and comparison.

---

> ### Author Response · Authors · 2022-08-02
> **Point-to-point Response to Reviewer iDDi**
>
> We sincerely appreciate your careful review and a great summary of our contributions. And thank you very much for the very constructive comments. In what follows, please see our responses.
>
> **Q1: It would be really helpful to add some discussion about the similarities and differences between the BLO and the L0-based pruning.**
>
> **A1:** This is an insightful question. In terms of similarity, both lines of research parameterize the pruning mask so that it is learnable compared to the heuristics-based pruning methods. The difference between BiP and L0-based pruning lies in the following aspects. First, BiP is not a sparse training algorithm since it calls a pre-trained model (see Line 282) as an initialization like IMP or OMP. This differs from L0-based pruning, which can also be modeled as a sparsity-inducing model training from scratch. See [R1] (Sec. 2) for a systematic classification of pruning methods.  Second, BiP and L0-based pruning enjoy quite different optimization foundations. Specifically, bi-level optimization (BLO) centered for BiP needs to tackle the challenge of implicit gradient (see Line 217 - 230 in our submission) due to the hierarchical learning structure of BLO. By contrast, the L0-based pruning is rooted in sparsity-inducing optimization [R2], which is typically formulated as a single-level minimization problem. However, we think the hierarchical learning structure is critical for pruning as this is also implied in IMP, the predominant pruning method to find “winning tickets”.  Third, BiP solves a constrained optimization problem, e.g., it calls the projected gradient descent for pruning. Yet, the L0-based pruning adopts a regularization scheme to strike a balance between performance and sparsity.
>
> > [R1] Liu, Shiwei, et al. "Sparse training via boosting pruning plasticity with neuroregeneration." Advances in Neural Information Processing Systems 34 (2021): 9908-9922.
> >
> > [R2] Bach, Francis, et al. "Optimization with sparsity-inducing penalties." Foundations and Trends® in Machine Learning 4.1 (2012): 1-106.
>
> **Q2: In Figure 6(b1) the best accuracy is achieved when lower-level steps N <= 3, which would be unexpected and need some explanation.**
>
> **A2:** In BiP, the (lower-level) $\theta$-step is initialized by a pre-trained model of high quality (test accuracy) already. Adopting the large lower-level step number (N > 3) will incur aggressive weight updating and lead to overfitting the current mask. In such a case, it is more difficult for the BLO solver to find a better mask. Therefore, the best performance is usually achieved at N <= 3.
>
> **Q3: Could you talk about the schedule of BIP to prune with multiple sparsity ratios?  When pruning for multiple ratios, the time complexity of IMP could be amortized and I wonder if BIP could be efficient in this setting.**
>
> **A3:** This is a very constructive comment. We agree that if multiple sparsity ratios are considered, then the time complexity of IMP could be amortized. Yet, it is worth noting that IMP typically calls for a strict sparsity schedule [R3] to achieve state-of-the-art performance (e.g., for the target pruning ratio of 51.2%, the schedule is 80% -> 64% -> 51.2% [R3]). Thus, IMP imposes “constraint” on the achieved “multiple sparsity ratios”. By contrast, our proposed BiP algorithm has no such constraint. In the worst case, one can call BiP multiple times to achieve multiple sparsity ratios. However, BiP could also be accelerated in this scenario using the low-sparsity solution as a warm-up to find the next higher-sparsity solution. This is a great direction to investigate: We will do it in the revised version.
>
> > [R3] Frankle, Jonathan, and Michael Carbin. "The lottery ticket hypothesis: Finding sparse, trainable neural networks." arXiv preprint arXiv:1803.03635 (2018).
>
> **Q4: It would be better to move some ablation studies like Figure A4 to the main manuscript.**
>
> **A4:** Thanks for your suggestion. We will move more ablation studies (Figure A4) to the main manuscript in the revised version.

---

> ### Author Response · Authors · 2022-08-08
> **Two days left for open discussion**
>
> Dear Reviewer iDDi:
>
> We are very grateful for your acknowledgment of our novelty and contributions. We have tried our best to address your questions. We list all the paper revisions in the **[summary of paper revisions and additional experiments](https://openreview.net/forum?id=t6O08FxvtBY&noteId=ckz79KyIMNi)**. As there are only two days left for author-reviewer discussions, we sincerely hope that you can provide us feedback before the discussion phase ends, and we are happy to answer any follow-up questions. Once again, thank you for your time and suggestions on our work.
>
> Best regards,
>
> Authors

---

### Official Review · Reviewer_c9rK · 2022-07-11

**Rating:** 7
**Confidence:** 4
**Soundness:** 4 excellent
**Presentation:** 4 excellent
**Contribution:** 4 excellent

**Summary:**

Search for the winning lottery in the Lottery Ticket Hypothesis (LTH) is of great interest in the Machine Learning (ML) community and this paper aims to find such winning tickets (in most cases). This work formulates the model pruning (primarily unstructured but also extends to structured pruning) as a Bilevel Optimization (BLO) where the lower-level optimization finds the best possible set of weights given the sparse neural network (that is weight masks) and the upper-level optimization is optimizing for the boolean mask (done using continuous relaxation followed by threshold-based rounding). Authors first derive the general expression for the gradient with respect to mask using an implicit gradient, which involves second-order derivatives, matrix inverse, and (n, n) matrices where n in #parameters in the network. However, with the hessian free assumption and given the nature of the problem (bilinear in mask “m” and parameters “theta”), the final expression of gradient turns out to have only first-order derivatives. Having defined the expressions, they finally proceed to describe the algorithm which involves lower-level SGD and upper-level SPGD, done until convergence, which does happen, in practice. Then the paper proceeds to the experiments involving multiple architectures, multiple datasets, and different pruning ratios. Performance metrics involve final accuracy, performance relative to the dense pre-trained model, and overall run time. Strong experiments show that BiP achieves superior performance than the original dense model (i.e. finds winning lottery) and has a comparable (or superior) performance to iterative magnitude pruning (IMP) while having much lesser run-time similar to one-shot pruning methods.

**Questions:**

As mentioned in the main review.

**Limitations:**

As mentioned in the main review.

**Strengths And Weaknesses:**

Strengths:

1. The Paper is concise and well written.
2. The theory is easy to follow and the experiment section is strong involving ablations across different hyperparameters.
3. Charts are well organized.

Overall, I enjoyed reading this work. I’ve some suggestions which primarily are additional references to be discussed and some additional visualization that could help to make this a strong submission which I describe in the following weakness section.

Weaknesses:
1. While the diagram is mentioned in the appendix, can the authors include an algorithm block for a better understanding of the overall flow?
2. How are the diverse batches chosen for the training? Does it involve some kind of submodular optimization to get the schedule? What happens if the batch for SGD is qualitatively different from the batch of SPGD?
3. The convergence result shown in the appendix involves final accuracy. Does this happen that the masks still keep on changing (that is finding the subnetwork) but all of that subnetworks have similar performance when trained (performed SGD)?
4. It was mentioned that SGD steps are kept fixed to 1. However at the very first step of pruning, why would still having just 1 SGD step suffice?
5. Table 1. Can authors add a vertical separator line for the datasets?

Additional references that can be discussed:
1. Soft Threshold Weight Reparameterization for Learnable Sparsity. ICML’20
2. EFFECTIVE TRAINING OF SPARSE NEURAL NETWORKS UNDER GLOBAL SPARSITY CONSTRAINT. CVPR’21
3. Rethinking Bi-Level Optimization in Neural Architecture Search: A Gibbs Sampling Perspective. AAAI’21

---

> ### Author Response · Authors · 2022-08-02
> **Point-to-point Response to Reviewer c9rK**
>
> We sincerely appreciate your careful review and a great summary of our contributions. Thank you very much for the very constructive comments. Please see our response below.
>
> **Q1: Can the authors include an algorithm block for a better understanding of the overall flow?**
>
> **A1:** Yes, we will add an algorithm block in the revision.
>
> **Q2: How are the diverse batches chosen for the training? Does it involve some kind of submodular optimization to get the schedule? What happens if the batch for SGD is qualitatively different from the batch of SPGD?**
>
> **A2:** The diverse batches for SGD and SPGD are realized by calling different rounds of random batch sampling from the data loader. We did not involve submodular optimization to get the schedule, but we think choosing qualitatively different batches is inspiring. For example, it might be an interesting future work to define a curriculum for training data used at upper-level and lower-level optimization, respectively. We will add this direction in Conclusion. Thanks for the comment.
>
> **Q3: Does the convergence result happen due to the fact that the masks still keep on changing (that is finding the subnetwork) but all of those subnetworks have similar performance when trained (performed SGD)?**
>
> **A3:** The mask also converges at the end of the training. To verify this argument, we show the training trajectory of the mask similarity between two adjacent-epoch models in [Figure](https://ibb.co/hcYQQX3) at different pruning ratios. Here the mask similarity is represented through the intersection of the union (IoU) score of the two masks found by two adjacent epochs. The IoU score ranges from 0.0 to 1.0, and a higher IoU implies a larger similarity between the two masks. As we can see, the IoU score converges to 1.0 in the end, which denotes that the mask also converges at the end of the training phase. Also, with a smaller pruning ratio, the mask turns to converge more quickly.
>
> **Q4: However, at the very first step of pruning, why would still having just 1 SGD step suffice?**
>
> **A4:** Thank you for raising this inspiring question. Based on the reviewer's comment, we conducted additional experiments to demonstrate the effectiveness of using one-step SGD in BiP. In our new experiments, we consider the number of SGD steps, 1, 3,  and 5. We report the training trajectories of BiP in [Figure](https://ibb.co/qYdpJHv). As we can see, the use of multi-step SGD accelerates model pruning convergence at its early phase. Yet, if we run BiP for a sufficient number of epochs (we used 100), the final test accuracy of using different SGD settings shows little difference. Although the use of multiple SGD steps could improve the convergence speed, it introduces extra computation complexity per BLO (bi-level optimization) step. Thus, from the overall computation complexity perspective, using 1 SGD step (even running for more epochs) is advantageous in practice.
>
> **Q5: Can authors add a vertical separator line for the datasets in Table 1?**
>
> **A5:** Yes, we will add a vertical separator line for the datasets in Table 1 in the revision.
>
> **Q6: Additional references that can be discussed**
>
> **A6:** Thank you for pointing out the additional related works. We will discuss the suggested related works in the revised version of our paper.
> 1. Soft Threshold Weight Reparameterization for Learnable Sparsity, ICML’20. The referred work developed an optimization-based pruning method that alternatively optimizes the parameters and the pruning thresholds. However, STR does not study the relationship between these two terms through the lens of bi-level optimization.
> 2. Effective Sparsification of Neural Networks with Global Sparsity Constraint, CVPR’21. This also provided an optimization-based pruning method that alternatively optimizes the model parameters and their corresponding pruning probabilities. Yet, we kindly stress that our proposed BiP algorithm is built on BLO, which, in contrast to the ordinary alternating optimization, requires an in-depth analysis of implicit gradients (see Line 217 - 230 in our submission).
> 3. Rethinking Bi-Level Optimization in Neural Architecture Search: A Gibbs Sampling Perspective, AAAI’21. This work considered optimizing the model architecture and parameters in a bi-level formulation, similar to DARTS. However, our proposed BLO for pruning is quite different from the previous DARTS-alike methods, since we show that the bi-linear nature of pruning variables gives a very special class of BLO problems that can be solved as easily as first-order optimization (see Line 252-254).
>
> We stress that there exist other methods using non-BLO alternating optimization-based pruning schemes. In contrast to our work, these methods neglect the role of implicit gradient (IG) Eq. (2) imposed by BLO. In our ablation study (Figure 6(b3)), we have demonstrated the necessity of the IG enhancement to fully exploit the potential of BLO to improve the performance of model pruning.

---

> > ### Comment · Reviewer_c9rK · 2022-08-05
> > **Thanks for the detailed and thorough responses**
> >
> > Most of my raised concerns are adequately satisfied, and I would indeed want to see future works that try with diverse batches, and study if it affects the convergence (either positively or negatively).

---

> > > ### Author Response · Authors · 2022-08-06
> > > **Thank you for your prompt response and encouragement!**
> > >
> > > We are happy to learn that our response has addressed most of your concerns adequately. Per reviewer’s encouragement, we would like to make a preliminary study to see how BiP performs if the upper-level data batch is in the reverse order of the lower-level data batch. We will update this response once we have these results.

---

> > > ### Author Response · Authors · 2022-08-06
> > > **Further response**
> > >
> > > Inspired by the reviewer’s follow-up comments, we conducted some additional experiments to further study the influence of the different upper- and lower-level data batch schemes on the pruning accuracy of BiP (**[Figure](https://ibb.co/hZhJR0P)**) as well as its convergence (**[Figure](https://ibb.co/ggkfxjD)**). We consider two variants of BiP using  different data batch schemes, termed BiP (reverse batch) and BiP (same batch), respectively. For BiP with reverse batch (a special mismatch case), the data batches from the same data loader are distributed to the upper-level SPGD and the lower-level SGD in the reverse order per epoch. For BiP with the same batch, the upper- and lower-level always adopt  the same data batch throughout the training. By default, BiP refers to our current implementation that uses different random data batches at two levels.
> > >
> > > As the results of BiP (reverse order) and BiP suggest, data batch mismatch is beneficial to BiP.  Even in the deterministic reverse order setting, the superior accuracy performance of BiP remains (see **[Figure](https://ibb.co/hZhJR0P)**). And its convergence behavior is similar to BiP (our current implementation) and outperforms BiP (same batch) (see **[Figure](https://ibb.co/ggkfxjD)**). We feel that it is a promising research direction to seek the optimal data batch curriculum for BiP’s lower-level and upper-level optimization in the future.
> > >
> > > Thanks again for this great comment.

---

> ### Author Response · Authors · 2022-08-08
> **Two days left for open discussion**
>
> Dear Reviewer c9rK:
>
> We are very grateful to your valuable comments. Thank you again for your prompt response. We have made a substantial effort in responding to your questions and have made an additional experiment inspired by your comment. The results and discussions can be found in our **[further response](https://openreview.net/forum?id=t6O08FxvtBY&noteId=1ClJcuzWCYS)**. We also list all the paper revisions and additional experiments in the **[summary of paper revisions and additional experiments](https://openreview.net/forum?id=t6O08FxvtBY&noteId=ckz79KyIMNi)**. As only two days are left for author-reviewer discussions, we are happy to answer any follow-up questions.
>
> Best regards,
>
> Authors

---

### Official Review · Reviewer_cu5X · 2022-07-16

**Rating:** 5
**Confidence:** 3
**Soundness:** 2 fair
**Presentation:** 3 good
**Contribution:** 2 fair

**Summary:**

This paper aims to solve unstructured pruning as a bi-level optimization problem. To find pruning masks and values of unpruned weights, they define bi-level equations and perform two SGD-processes iteratively. Their goal is proposing a new method that can show comparable accuracy as IMP(iterative magnitude-based pruning) with restricted training time.

**Questions:**

- How about bringing the Appendix B (including Figure A1) to the main manuscript? At the first time, it was a little hard to understand the overview of this method.
- I don’t understand why LTH is referred as  a pruning method. As far as I understand, the LTH paper just used the magnitude-based pruning method for finding winning-tickets. LTH paper is a novel paper to represent why we train the over-parameterized neural networks, not represent a new pruning method. I think IMP with gradual pruning rates was proposed in Zhu’s paper (described below).
Why are the below papers not mentioned in this paper? There have been many other results on unstructured pruning methods.
  - Zhu, Michael, and Suyog Gupta. "To prune, or not to prune: exploring the efficacy of pruning for model compression." arXiv preprint arXiv:1710.01878 (2017).
  - Evci, Utku, et al. "Rigging the lottery: Making all tickets winners." International Conference on Machine Learning. PMLR, 2020.
Sidak Pal Singh and Dan Alistarh. Woodfisher: Efficient second-order approximation for neural network compression. Conference on Neural Information Processing Systems (NeurIPS), 33, 2020.
  - Peste, Alexandra, et al. "Ac/dc: Alternating compressed/decompressed training of deep neural networks." Advances in Neural Information Processing Systems 34 (2021): 8557-8570.


**Limitations:**

- This paper argues for an unstructured pruning method, but there is a critical issue on the unstructured pruning. I think this paper has no/less consideration on that.
- The experimental designs are so restricted. It should be extended to other challenging datasets and architectures.


**Strengths And Weaknesses:**

### Strengths
- For unstructured pruning of DNNs, this paper defines a bi-level optimization problem and shows theoretical contents for that.
This paper shows the comparable results of this method using various dataset and architectures.
- The compression (including re-training) time is not increased with various pruning rates.

### Weaknesses
- I have a concern on the real effectiveness of unstructured pruning. As many papers have been mentioned, there is no realistic acceleration method for unstructured pruning due to random locations of unpruned weights. There is no note for this widely-known problem. Even though this method is a novel idea for unstructured pruning, we cannot use this method for our inference. When using CSR formats (ex. cusparse library in CUDA), we must need higher pruning rates to gain faster inference speed. I think that problem is why there have been a few studies on unstructured pruning these days unlike structured pruning.
- In this aspect, this method can achieve outperformed results on lower pruning rates less than 50%. But, there is less noticeable improvement on effective pruning rates. So, in my opinion, the resulting accuracy seems to be not the main contribution of this paper.
Moreover, the experimental results seem to be so restricted. We are living in 2022 and the neural networks have evolved since Song Han’s magnitude-based pruning. I don’t think the results on the CIFAR-10 dataset can prove the novelty of a pruning method. It can just show a pruning method works well. Most of the results on this paper are limited to the ResNet arch. and CIFAR-10 dataset. CIFAR-10 consists of just ten classes. I think this paper should extend the experimental results to various architectures (ex. MobileNet, Transformer, …) or bigger dataset with smaller model (ex. ResNet-18 on ImageNet).
- Then, there remains another main contribution, faster pruning-retraining speed regardless of pruning rate. To strengthen this contribution, I think there should be more ablation studies and experiments. I’m curious why this method works with consistent time regardless of pruning rates. Or if I put much time for higher pruning rates, then can I gather higher accuracy by using this method?

---

> ### Author Response · Authors · 2022-08-02
> **Point-to-point Response to Reviewer cu5X (Part 3/3)**
>
> **Q5: How about bringing the Appendix B (including Figure A1) to the main manuscript?**
>
> **A5:** Thanks for your suggestion. We will add the content of Appendix B to the main manuscript in the revised version.
>
> **Q6: I don’t understand why LTH is referred as a pruning method. LTH paper is a novel paper to represent why we train the over-parameterized neural networks, not represent a new pruning method. I think IMP with gradual pruning rates was proposed in Zhu’s paper. There have been many other results on unstructured pruning methods.**
>
> **A6:** Yes, we agree with the reviewer that IMP is the name of the pruning method that we should call. We have avoided the use of ‘LTH pruning’ in the original submission and will carefully check our statement to make it preciser in the revision.  Meanwhile, we would like to bring to the reviewer’s attention that the LTH paper and its follow-up work made some modifications/customizations to the vanilla IMP approach, e.g., the use of initialization rewinding at every pruning round [R6, R7, R8]. Thus, when we refer to IMP, it represents the IMP algorithm used to find the best winning tickets in the line of LTH research [R7, R8] rather than the one used in Zhu’s paper [R10]. Sorry for this confusion, and we will cite [R10] and clarify its difference with the LTH work in the revision.
>
> Thank you very much for pointing out the additional related work [R10-R13]. We will be sure to cite them and state their differences from ours in the revised paper. Meanwhile, we would like to kindly stress that different from these work, our paper aims to re-think the optimization basis of network pruning through the lens of BLO and to develop a theoretically-grounded and effective BLO solver for model pruning that can attain high pruned model accuracy and high computational efficiency.
>
> > [R6] Frankle, Jonathan, and Michael Carbin. "The lottery ticket hypothesis: Finding sparse, trainable neural networks." arXiv preprint arXiv:1803.03635 (2018).
> >
> > [R7] Frankle, Jonathan, David J. Schwab, and Ari S. Morcos. “The early phase of neural network training.” arXiv preprint arXiv:2002.10365 (2020).
> >
> > [R8] Renda, Alex, Jonathan Frankle, and Michael Carbin. "Comparing rewinding and fine-tuning in neural network pruning." arXiv preprint arXiv:2003.02389 (2020).
> >
> > [R9] Ma, Xiaolong, et al. “Sanity checks for lottery tickets: Does your winning ticket really win the jackpot?.” Advances in Neural Information Processing Systems 34 (2021): 12749-12760.
> >
> > [R10] Zhu, Michael, and Suyog Gupta. “To prune, or not to prune: exploring the efficacy of pruning for model compression.” arXiv preprint arXiv:1710.01878 (2017).
> >
> > [R11] Evci, Utku, et al. "Rigging the lottery: Making all tickets winners." International Conference on Machine Learning. PMLR, 2020.
> >
> > [R12] Singh, Sidak Pal, and Dan Alistarh. "Woodfisher: Efficient second-order approximation for neural network compression." Advances in Neural Information Processing Systems 33 (2020): 18098-18109.
> >
> > [R13] Peste, Alexandra, et al. "Ac/dc: Alternating compressed/decompressed training of deep neural networks." Advances in Neural Information Processing Systems 34 (2021): 8557-8570.

---

> ### Author Response · Authors · 2022-08-02
> **Point-to-point Response to Reviewer cu5X (Part 2/3)**
>
> **Q3: The experimental results seem to be so restricted and most of the results on this paper are limited to the ResNet arch. and CIFAR-10 dataset. I think this paper should extend the experimental results to various architectures (ex. MobileNet, Transformer, …) or bigger dataset with smaller model (ex. ResNet-18 on ImageNet). The experimental designs are so restricted. It should be extended to other challenging datasets and architectures.**
>
> **A3:** We respectfully disagree that our experiments are restricted. We have followed the experiment setup in the latest pruning benchmark [R4] to make sure that our improvement is consistent and solid across different settings. Reviewer c9rK also recognized the  strength of our experiments in her/his comment “the experiment section is strong involving ablations across different hyperparameters.”  To be specific, our experiment plans include **four datasets** (CIFAR-10, CIFAR-100, Tiny-ImageNet (200 classes), and ImageNet (1000 classes)), **five model architectures** (ResNet-20, ResNet-56, ResNet-18, ResNet-50, and VGG-16), and **three pruning settings** (unstructured pruning, filter-wise structured pruning as well as channel-wise structured pruning); see a summary of our experiment setting in Sec. 4.1 and some results in Tab. 1, Figure 3, 4, A6, and A7.
>
> Thank you for suggesting extending the experimental results to the other architectures (e.g., MobileNet, and Transformer) and ResNet-18 on ImageNet. We plan to include the new experiment on (ResNet-18, ImageNet) as a supplement to (ResNet-50, ImageNet) in Fig. 3. We will report the results once this experiment is finished. If time is allowed, we will add the suggested experiment on MobileNet. Yet, we feel that pruning Transformer could be quite different from pruning CNN-type networks; E.g., token sparsification is a key step of transformer pruning [R5]. We believe this is outside the scope of our work. Thus, we will leave this study for future research.
>
> > [R4] Ma, Xiaolong, et al. “Sanity checks for lottery tickets: Does your winning ticket really win the jackpot?.” Advances in Neural Information Processing Systems 34 (2021): 12749-12760.
> >
> > [R5] Rao, Yongming, et al. "DynamicViT: Efficient vision transformers with dynamic token sparsification." Advances in neural information processing systems 34 (2021): 13937-13949.
>
> **Q4: I’m curious why this method works with consistent time regardless of pruning rates. Or if I put much time for higher pruning rates, then can I gather higher accuracy by using this method?**
>
> **A4:** The reason for constant time regardless of pruning rates is that the proposed optimization scheme is independent of the specific value of the pruning ratio. This differs from IMP, where the number of pruning rounds increases as the pruning rate increases. In BiP (Eq. (1)), the pruning ratio is regarded as an upper-level constraint. Thus, even if a higher pruning rate is applied, the only change made to BiP is using a different projection threshold in the upper-level SPGD; see (m-step) at Line 261. This does not require the increase of optimization steps.
>
> Based on the reviewer’s suggestion, we also conducted new experiments to allow more time (training epochs) for BiP when a higher pruning rate is considered; see results in [Figure](https://ibb.co/SfWD49m). Specifically, we test three datasets and consider three pruning ratios (p=86.58%, 94.50%, 97.75%). For each pruning ratio, we examine the test accuracy of BiP versus the training epoch number from 50 to 500. Note that the number of training epochs in our original experiment setup was 100. As we can see, the performance of BiP gets saturated when the epoch number is over 100. Thus, increasing the training epoch number (over 100) does not improve accuracy at a higher pruning ratio.

---

> ### Author Response · Authors · 2022-08-02
> **Point-to-point Response to Reviewer cu5X (Part 1/3)**
>
> We thank the reviewer for the detailed feedback. Please find our detailed responses below.
>
> **Q1: I have a concern on the real effectiveness of unstructured pruning although this method is a novel idea for unstructured pruning.**
>
> **A1:** This is a great comment. First of all,  we agree that unstructured pruning may not result in direct acceleration, and it is the reason for considering structured pruning for the actual model deployment. However, this does NOT restrict the novelty of our work. Yes, we considered unstructured pruning as the representative use case, but the proposed BLO (bi-level optimization)-oriented pruning algorithm (that we call BiP) is generic and can be readily extended to the use case of structured pruning. This is also one of our novelties. We have shown the superior performance of BiP in two structured pruning settings, namely, filter pruning and channel pruning across various datasets (see results in Figure 4, A5, A6, A7, and Lines 298-301, 337-343). We would like to kindly stress that the main goal of our work is to re-think the optimization basis of network pruning through the lens of BLO and to attain high pruned model accuracy (like IMP) and high computational efficiency (like OMP).
>
> Secondly, although unstructured pruning cannot bring in realistic acceleration, it still serves as a technology basis for sparse learning of deep neural networks (DNNs). Besides efficiency, many other metrics are strongly related to sparsity, e.g., adversarial robustness [R1], out-of-distribution generalization [R2], and model transferability [R3].  All the aforementioned work focused on unstructured pruning. Thus, unstructured pruning is still an important topic to study. We will surely add the above discussion in the revision to improve our motivation for unstructured pruning. Thank you for your great comment!
>
> > [R1] Sehwag, Vikash, et al. "Hydra: Pruning adversarially robust neural networks." Advances in Neural Information Processing Systems 33 (2020): 19655-19666.
> >
> > [R2] Diffenderfer, James, et al. "A winning hand: Compressing deep networks can improve out-of-distribution robustness." Advances in Neural Information Processing Systems 34 (2021): 664-676.
> >
> > [R3] Chen, Tianlong, et al. "The lottery tickets hypothesis for supervised and self-supervised pre-training in computer vision models." Proceedings of the IEEE/CVF Conference on Computer Vision and Pattern Recognition. 2021.
>
> **Q2: This method can achieve outperformed results on lower pruning rates less than 50%. But, there is less noticeable improvement on effective pruning rates. So, in my opinion, the resulting accuracy seems to be not the main contribution of this paper.**
>
> **A2:** At the first glance, the most significant accuracy improvement of BiP over baselines stays in the sparse regime of less than 50%. Yet, this is not precise. First, our results in Tab. 1 have shown that BiP consistently identifies a ‘winning’ subnetwork with the highest sparsity level compared to the other baseline methods (see Line 333-335). Second, we agree that the effectiveness of pruning inevitably drops as the network becomes increasingly sparser. Thus, how to improve the effectiveness of pruning in the extremely-sparse regime is an open challenge. Third, we have shown that BiP consistently improves accuracy in both unstructured and structured pruning scenarios. Based on the above, we believe that the accuracy performance of BiP is still the main contribution of this paper.

---

> ### Author Response · Authors · 2022-08-06
> **Look forward to post-rebuttal feedback!**
>
> Dear reviewer cu5X,
>
> Thank you very much for taking the time to review our paper. We cherish your comments very much. In our earlier posted responses, we have made a point-to-point response (see [Part I](https://openreview.net/forum?id=t6O08FxvtBY&noteId=jBWSEYbF4ke), [Part II](https://openreview.net/forum?id=t6O08FxvtBY&noteId=56J1_z-Dcb), [Part III](https://openreview.net/forum?id=t6O08FxvtBY&noteId=VE5Xepxej7f) respectively) to alleviate your concern. If you have additional comments, we are happy to address them.
>
> Authors

---

> ### Author Response · Authors · 2022-08-08
> **Two days left for open discussion**
>
> Dear Reviewer cu5X:
>
> We are very grateful to your constructive comments. We have made a substantial effort in responding to your questions and listed all the paper revisions and additional experiments in the **[summary of paper revisions and additional experiments](https://openreview.net/forum?id=t6O08FxvtBY&noteId=ckz79KyIMNi)**. As there are only two days left for author-reviewer discussions, we sincerely hope that you could provide us feedback before the discussion phase ends, and are happy to answer any follow-up questions.
>
> Best regards,
>
> Authors

---

> > ### Comment · Reviewer_cu5X · 2022-08-08
> > **Thank you for detailed responses and revisions.**
> >
> > I would like to thank the authors' feedback and revisions in the manuscript, which could solve my several concerns. But, I still want to keep my initial score because I don’t agree that the experiments are enough to prove this pruning method works well for general architectures. The models have been known as highly redundant models. Unfortunately, I strongly think we need another baseline except for CIFAR-10. At least, I think this paper should include many results by using ImageNet or other tasks. For Table 1, I also keep my opinion on CIFAR-10/100 datasets. I hope I could find this paper as a refined version with various and challenging architectures/dataset in near future, but I also respect other reviewers’ opinions.

---

> > > ### Author Response · Authors · 2022-08-08
> > > **Thank you and further response**
> > >
> > > Dear reviewer cu5X:
> > >
> > > We are glad to see that our previous responses have addressed some of your concerns. For your remaining concern, please allow us to provide the following responses.
> > >
> > > Following your comments, we have been running additional experiments on (ResNet18, ImageNet) and (MobileNet, TinyImageNet).
> > >
> > > We just finished the experiments on (ResNet18, ImageNet) and show our results in **[Figure](https://ibb.co/9HcGbs1)** (Figure A17 in our revision). As we can see, consistent with the existing results on (ResNet50, ImageNet) in Figure 3, our proposed BiP method remains superior to the IMP baseline. We would also like to mention that pruning on ImageNet over multiple sparse ratios is very time-consuming (especially for IMP). Conducting the suggested experiment has taken 12 V100 GPUs (nearly all of our computing resources). We hope that the reviewer can see our effort in addressing your comment.
> > >
> > > We are running another experiment on (MobileNet, TinyImageNet) and will update this response once the results are ready.
> > >
> > > We kindly point out that although Table 1 only contains CIFAR-10/100 results, it can readily be extended to cover our existing results on **TinyImageNet and ImageNet**. This is because this table is associated with Figure 3 but provides a clearer description of the accuracy of the winning ticket vs. its highest pruning ratio. To make this point clearer, we provide the enriched Table 1 in **[Figure](https://ibb.co/DgWFv94)** and Table A4 in the revision (we did not add the additional results to Table 1 due to the space limitation).
> > >
> > > It is a great comment to consider less redundant models. Yet, we kindly bring the reviewer’s attention to two aspects.
> > > **First**, we have considered less redundant model architectures ResNet20 and ResNet56, which have only 0.27M and 0.85M parameters and much less than ResNet18 (11M) (see Table A1). As shown in Fig. 3 and 4, our method consistently outperforms the baselines, although less redundant models would be more difficult to prune. **Second**, when we choose model architectures for various datasets, we stick to the dense models widely used in the pruning literature, e.g., the NeurIPS’21 benchmark [R1]. We can achieve state-of-the-art accuracy on each dataset (note that the less redundant models typically correspond to lower test accuracy). **Third**, compared to experiments in many recent pruning works [R1-R5], the architectures and datasets considered in our work are indeed comprehensive: We kindly bring reviewers' attention to our results in the appendix, e.g., Figs A6, A7, A9, A10 besides Figure 3 and 4 in the main texts. By contrast, [R2, R3] did not consider VGG16, and [R1, R4, R5] did not consider ResNet56.
> > >
> > > In summary, we highly appreciate the reviewer's effort in providing many useful comments on our submission. Based on our existing experiments (in both the main paper and the supplement) and the newly-added experiments (in both the first-round response and the current response), we sincerely hope that the remaining concern has been alleviated and you could be open to adjusting your rating.
> > >
> > > Thank you very much,
> > >
> > >
> > >
> > > > [R1] Ma, Xiaolong, et al. “Sanity checks for lottery tickets: Does your winning ticket really win the jackpot?.” Advances in Neural Information Processing Systems 34 (2021): 12749-12760.
> > > >
> > > > [R2] Singh, Sidak Pal, and Dan Alistarh. "Woodfisher: Efficient second-order approximation for neural network compression." Advances in Neural Information Processing Systems 33 (2020): 18098-18109.
> > > >
> > > > [R3] Evci, Utku, et al. "Rigging the lottery: Making all tickets winners." International Conference on Machine Learning. PMLR, 2020.
> > > >
> > > > [R4] Peste, Alexandra, et al. "Ac/dc: Alternating compressed/decompressed training of deep neural networks." Advances in Neural Information Processing Systems 34 (2021): 8557-8570.
> > > >
> > > > [R5] Alizadeh, Milad, et al. "Prospect pruning: Finding trainable weights at initialization using meta-gradients." arXiv preprint arXiv:2202.08132 (2022).

---

### Author Response · Authors · 2022-08-08
**Summary of Paper Revisions and Additional Experiments**

Dear reviewers,

We are glad to receive your valuable and constructive comments. We have made a substantial effort to clarify your doubts and enrich our experiments in the rebuttal phase. Below is a summary of revisions:

Changes suggested by the reviewers:
- (Reviewer cu5X): (1) We discuss the importance of unstructured pruning; see Line 30\~32. (2) We add related work on unstructured pruning; see Line 94. (3) Following Table 1, we add the results for Tiny-ImageNet and ImageNet to Table A4.

- (Reviewer c9rK): (1) We add a vertical separator line for the datasets in Table1. (2) We add related work on optimization-based pruning; see Line 98.

- (Reviewer mQEK): (1) We discuss “Differentiable Network Pruning for Microcontrollers” and talk about our paper’s novelty on bi-level optimization; see Line 162\~165. (2) We clarify the definition of the winning ticket; see Line 104\~108.

- (Reviewer iDDi): (1) We add related work on L0-based pruning; see Line 97.

- (All reviewers): Due to paper limitations, we defer the major revisions such as moving ablation studies and algorithm blocks to the main texts in the next version. Thank you again for the suggestions.

Below is a summary of the additional results:

- Reviewer [cu5X](https://openreview.net/forum?id=t6O08FxvtBY&noteId=cQiBjp_bI0P):
    1. We conducted new experiments to investigate the performance of BiP if more training epochs are used at a higher pruning rate; see results in **[Figure](https://ibb.co/SfWD49m)** and analysis in [Q4&A4](https://openreview.net/forum?id=t6O08FxvtBY&noteId=56J1_z-Dcb). The results are also included in the revised manuscript (Figure A12).
    2. We conducted new experiments on (ResNet18, ImageNet); see results in **[Figure](https://ibb.co/9HcGbs1)**. The results are also included in the revised manuscript (Figure A17).
- Reviewer [c9rK](https://openreview.net/forum?id=t6O08FxvtBY&noteId=jzC7z6ztw2o):
    1. We verified the convergence of pruning masks by tracking the IoU score between the masks at two adjacent epochs; see results in **[Figure](https://imgbb.com/hcYQQX3)** and analysis in [Q3&A3](https://openreview.net/forum?id=t6O08FxvtBY&noteId=jzC7z6ztw2o). The results are also included in the revised manuscript (Figure A14).
    2. We showed the training dynamics of BiP using different lower-level SGD steps and demonstrated the effectiveness of using one-step SGD in the lower level of BiP; see results in **[Figure](https://imgbb.com/qYdpJHv)** and analysis in [Q4&A4](https://openreview.net/forum?id=t6O08FxvtBY&noteId=jzC7z6ztw2o). The results are also included in the revised manuscript (Figure A13).
    3. We showed the influence of the different upper- and lower-level data batch schemes on the pruning accuracy of BiP; see results in **[Figure](https://ibb.co/hZhJR0P)** and analysis in [Further Response](https://openreview.net/forum?id=t6O08FxvtBY&noteId=1ClJcuzWCYS). The results are also included in the revised manuscript (Figure A16).
- Reviewer [mQEK](https://openreview.net/forum?id=t6O08FxvtBY&noteId=D-YYStKy9FM):
    1. For unstructured pruning settings, we add the baselines Early-Bird and ProsPr under CIFAR-10, CIFAR-100, and TinyImageNet datasets (6 model architecture + dataset combinations). The results can be found in **[Figure](https://ibb.co/TbqXNFc)**. A detailed discussion can be found in [Q3&A3](https://openreview.net/forum?id=t6O08FxvtBY&noteId=axWLRoemTAz). The results are also included in the revised manuscript (Figure A9).
    2. For structured pruning settings, we add ProsPr as the latest initialization-based baseline (4 model architecture + dataset combinations). The results can be found in **[Figure](https://ibb.co/K0PTPnP)**. A detailed discussion can be found in [Q7&A7](https://openreview.net/forum?id=t6O08FxvtBY&noteId=axWLRoemTAz). The results are also included in the revised manuscript (Figure A10).
    3. To verify the insensitivity of BiP to rewinding epochs, we conducted additional experiments on 3 more dataset-model architecture combinations (ResNet18 + CIFAR-10, ResNet18 + CIFAR-100, ResNet18 + TinyImageNet). The results can be found in **[Figure](https://ibb.co/Dzf5YQN)**. More detailed discussions can be found in [Q10&A10](https://openreview.net/forum?id=t6O08FxvtBY&noteId=_HzCMw77WGT). The results are also included in the revised manuscript (Figure A11).

---

### Meta-Review · Area_Chair_vmLx · 2022-08-30

**Recommendation:** Accept
**Confidence:** Less certain

**Metareview:**

The reviewers had significantly diverging opinions on this manuscript. The main issue under discussion was whether the framing of this paper as a lottery ticket work was correct, given that the main evaluations use no reinitialization or rewinding. On balance, I think that while one reviewer was very negative about the paper, the disagreement was mostly terminological. The substantial concern is whether the evaluation comparison (wherein BLO with no rewinding is compared against methods that use rewinding in Figure 3) is fair. The authors respond to this by providing comparisons in Figures 6A and A11 that evaluate rewinding on some tasks. However, these figures seem to show that the accuracy of BLO is completely insensitive to rewinding—and even to complete reinitialization in the original lottery-ticket sense. This raises the natural question: why not just evaluate primarily in the reinitialized case, where there's no need to redefine the term "winning ticket"? That is, the whole presentation seems to be backwards. The way it should be presented/evaluated is:

* First, we show that BLO outperforms other methods in the classic lottery ticket regime, where we reset all the weights to initialization (100% rewind) — this would replace the present Figure 3. This would be a fair comparison, comparing classical-lottery-ticket-setting pruning methods to each other.
* Next, we show that one advantage of BLO is that unlike other winning-ticket-finding methods, its performance is invariant to rewinding. That is, if what we want is just accuracy of the pruned model (and not to do some sort of scientific investigation of the lottery ticket hypothesis) then BLO outperforms other methods when we don't do any rewinding at all.

With this sort of presentation, I think the authors could have avoided the negative reviewer's objections. Despite these presentational/terminological issues, I think that there's enough technical contribution here with Figures 6A and A11 to move forward with acceptance, especially considering the enthusiasm of the other reviewers and the technical novelty of the bi-level approach. The empirical results _are_ there (and Figures 6A and A11 show a clear connection with the lottery ticket work), they're just presented strangely. And I think there are not any fundamental technical issues here that forbid acceptance.

**Award:**

No

---

### Decision · Program_Chairs · 2022-09-14

Accept